# Pervasive coexpression of spatially proximal genes is buffered at the protein level

Georg Kustatscher[1],[*] iD, Piotr Grabowski[2] iD & Juri Rappsilber[1,2],[**] iD

## Abstract

Genes are not randomly distributed in the genome. In humans, 10% of protein-coding genes are transcribed from bidirectional promoters and many more are organised in larger clusters. Intriguingly, neighbouring genes are frequently coexpressed but rarely functionally related. Here we show that coexpression of bidirectional gene pairs, and closeby genes in general, is buffered at the protein level. Taking into account the 3D architecture of the genome, we find that co-regulation of spatially close, functionally unrelated genes is pervasive at the transcriptome level, but does not extend to the proteome. We present evidence that non-functional mRNA coexpression in human cells arises from stochastic chromatin fluctuations and direct regulatory interference between spatially close genes. Protein-level buffering likely reflects a lack of coordination of post-transcriptional regulation of functionally unrelated genes. Grouping human genes together along the genome sequence, or through long-range chromosome folding, is associated with reduced expression noise. Our results support the hypothesis that the selection for noise reduction is a major driver of the evolution of genome organisation.

**Keywords** gene expression noise; genome organisation; proteomics; regulatory interference; transcriptomics
**Subject Categories** Chromatin, Epigenetics, Genomics & Functional Genomics; Genome-Scale & Integrative Biology
**Mol Syst Biol. (2017) 13: 937**

## Introduction

The position of genes in the human genome is not random (Hurst *et al*, 2004). Genes are often found in pairs or larger clusters that tend to be coexpressed (Caron *et al*, 2001; Lercher *et al*, 2002; Trinklein *et al*, 2004). Some of these coordinate transcription of genes with related functions, for example histone genes and other clusters resulting from gene duplication. However, the majority of closeby, coexpressed human genes appear not to have a higher functional similarity than random gene pairs (Hurst *et al*, 2004; Williams & Bowles, 2004; Li *et al*, 2006; Purmann *et al*, 2007; Michalak, 2008; Xu *et al*, 2012). For example, 35 DNA repair genes are transcribed from bidirectional promoters, but none of their paired genes is involved in DNA repair (Xu *et al*, 2012). This raises intriguing questions: Why are functionally unrelated genes clustered in the genome and how can the cell tolerate their coexpression?

Pioneering work in yeast identified the selection for reduced gene expression noise as a key driver for the evolution of chromosome organisation (Batada & Hurst, 2007; Wang *et al*, 2011). A major cause of gene expression noise is thought to be the random fluctuation of chromatin domains between an active and inactive state, causing mRNAs to be synthesised in short, stochastic bursts (Raj *et al*, 2006). Clusters of active genes may mutually reinforce their open chromatin state, minimising stochastic chromatin remodelling, and thereby reduce expression noise (Batada & Hurst, 2007; Wang *et al*, 2011). Similarly, genes flanking bidirectional promoters have lower expression noise than other genes, even if one of the divergent partners is a noncoding RNA (Wang *et al*, 2011). Noise-sensitive genes, such as those encoding protein complex subunits, are enriched among bidirectional pairs, but neither in yeast nor in human do any of these pairs encode two subunits of the same protein complex (Li *et al*, 2006; Wang *et al*, 2011). Consequently, it has been suggested that bidirectional promoters may drive noise reduction rather than the coexpression of functionally related genes (Wang *et al*, 2011).

The noise reduction model not only provides a potential explanation for the occurrence of clusters of functionally unrelated genes, but also predicts that such genes may be coexpressed (Wang *et al*, 2011). In yeast, chromatin-modifying enzymes are major contributors to gene expression noise (Newman *et al*, 2006) and chromatin remodelling drives the incidental coexpression of neighbouring, functionally unrelated genes (Batada *et al*, 2007). This coexpression may be due to a passive mechanism, whereby random transitions between open and closed chromatin simultaneously expose all genes within a chromatin domain to the transcriptional machinery. Alternatively, for very close genes such as those with bidirectional promoters, the up- or downregulation of one gene may directly affect the transcriptional status of its neighbour (Wang *et al*, 2011). Indeed, such a "ripple effect" of transcriptional activation has been observed in yeast and humans (Ebisuya *et al*, 2008). The noise and

1 Wellcome Trust Centre for Cell Biology, University of Edinburgh, Edinburgh, UK
2 Chair of Bioanalytics, Institute of Biotechnology, Technische Universität Berlin, Berlin, Germany
*Corresponding author. Tel: +44 131 6517057; E-mail: georg.kustatscher@ed.ac.uk
**Corresponding author. Tel: +44 131 6517056; E-mail: juri.rappsilber@ed.ac.uk

expression levels of transgenes also vary with their insertion site, as a result of both domain-wide effects and interference with individual neighbouring genes (Gierman *et al*, 2007; Chen & Zhang, 2016). Transgenes can also affect the mRNA expression levels of endogenous genes located close to the insertion site (Akhtar *et al*, 2013).

If the transcription of noise-reduced, clustered genes is unduly influenced by their neighbours, how can individual genes reach their optimal expression levels? Notably, gene expression is usually measured at the mRNA level. However, protein levels are buffered against certain transcript fluctuations (Liu *et al*, 2016), such as those caused by stochastic transcription initiation (Raj *et al*, 2006; Gandhi *et al*, 2011) and genetic variation between individuals (Battle *et al*, 2015) and species (Khan *et al*, 2013). The abundance of some proteins can also be buffered against gene copy number variations (Geiger *et al*, 2010; Stingele *et al*, 2012; Dephoure *et al*, 2014). We therefore speculated that protein abundances may also be buffered against regulatory interference between genes in close spatial proximity.

# Results

### Coexpression of bidirectional gene pairs is buffered at the protein level

We investigated the expression of 4,188 genes across 60 different human lymphoblastoid cell lines (LCLs), for which mRNA (Pickrell *et al*, 2010) and protein abundances (Battle *et al*, 2015) have been reported (Fig 1A, Dataset EV1). These genes are highly expressed in all human tissues and their promoters are in active chromatin states (Appendix Fig S1). Although constitutively active, expression levels of these "housekeeping" genes vary between LCLs, as a result of genetic and other differences, including age and growth conditions (Akey *et al*, 2007; Stark *et al*, 2014; Yuan *et al*, 2015). The LCL cell line panel has been instrumental in identifying expression quantitative trait loci, that is DNA sequence variants that specifically influence the expression level of one or more genes (Albert & Kruglyak, 2015). Here, instead of assessing how a gene's expression level depends on the genotype, we analyse how it is influenced by the expression of other, closeby genes. LCLs are a valuable test system as their genome structure and regulatory elements have been mapped at unparalleled resolution (Lieberman-Aiden *et al*, 2009; Ernst *et al*, 2011; ENCODE Project Consortium, 2012; Rao *et al*, 2014).

First, we analysed gene pairs that are transcribed from bidirectional promoters. These are commonly defined as genes that are found in head-to-head orientation with < 1 kb between their transcription start sites (TSSs) (Trinklein *et al*, 2004). Out of 167 such gene pairs in this dataset, the mRNA abundances of 31 (19%) are strongly and significantly co-regulated across LCLs (Pearson's correlation coefficient, PCC > 0.5, BH-adjusted *P*-value < 0.001). However, protein co-regulation is attenuated or buffered for 28 of these (Fig 1B, Appendix Table S1). Literature analysis revealed that the buffered gene pairs generally have unrelated biological functions, in contrast to the three gene pairs whose co-regulation is sustained at the protein level (Appendix Table S1).

We next considered the 929 non-bidirectional gene pairs with up to 50 kb between their TSSs, regardless of their orientation (Dataset EV2). Although these pairs do not share a promoter region, we find that 22% have co-regulated mRNA abundances (PCC > 0.5, BH-adjusted *P* < 0.001). However, only 3% are also co-regulated at the protein level (Fig 1B).

### Genes with similar functions have co-regulated mRNA and protein abundances

To confirm that the different impact of gene proximity on mRNA and protein abundances reflects a biological phenomenon, rather than simply a difference in data quality, we assessed the co-regulation of genes with known functional links, irrespective of their genomic position. We analysed subunits of the same protein complex, enzymes catalysing consecutive reactions in metabolic pathways and proteins with identical subcellular localisations. In all cases, we observe strong co-regulation on mRNA and protein levels, but co-regulation of proteins is significantly stronger than that of mRNAs (Fig EV1, $P < 3 \times 10^{-16}$). Therefore, data quality appears not to be limiting. Instead, the observed differences between mRNA and protein co-regulation indicate that post-transcriptional processes eliminate co-regulation of genes which are related spatially, but not functionally.

### A fraction of closeby genes is enriched for similar functions

Our observation that only 3% of closeby genes have co-regulated protein abundances appears to contrast with the fact that genes in close genomic proximity are enriched for similar functions (Thévenin *et al*, 2014). However, functional enrichment does not exclude the possibility that the bulk of closeby gene pairs does not share similar functions. For example, we find that co-regulation of transcripts and proteins from closeby genes is more common than for random protein pairs (Fig 1B), and this enrichment is highly significant (3% versus 0.4%, $P < 4 \times 10^{-14}$).

To analyse the relationship between gene distance and function more systematically, we assessed functional associations between our gene pairs using the STRING database (Szklarczyk *et al*, 2017). We considered gene pairs to be functionally associated if their STRING score, that is the likelihood of the association to be biologically meaningful, specific and reproducible, was > 0.7. Using this comprehensive definition, we find that 4.5% of closeby gene pairs, that is those with < 50 kb between their TSSs, are related functionally (Fig EV2A). As observed by Thévenin *et al*, we find this to be a significant enrichment over gene pairs that are farther apart. Likewise, gene pairs from the same chromosome are enriched for similar functions relative to those from different chromosomes. Nevertheless, the extent of mRNA co-regulation (22%) strongly exceeds co-function, and mRNA co-regulation of most closeby gene pairs is not sustained at the protein level (Fig EV2A).

Notably, a similar analysis in yeast has shown that adjacent genes tend to have correlated mRNA expression and are statistically enriched for similar functions (Cohen *et al*, 2000). However, in striking agreement with our observations, only about 2% of these coexpressed neighbouring gene pairs have related functions (Batada *et al*, 2007) and only for these is gene order evolutionarily conserved (Hurst *et al*, 2002). Coexpression of neighbouring genes has also been observed in *Arabidopsis thaliana*, but only a fraction

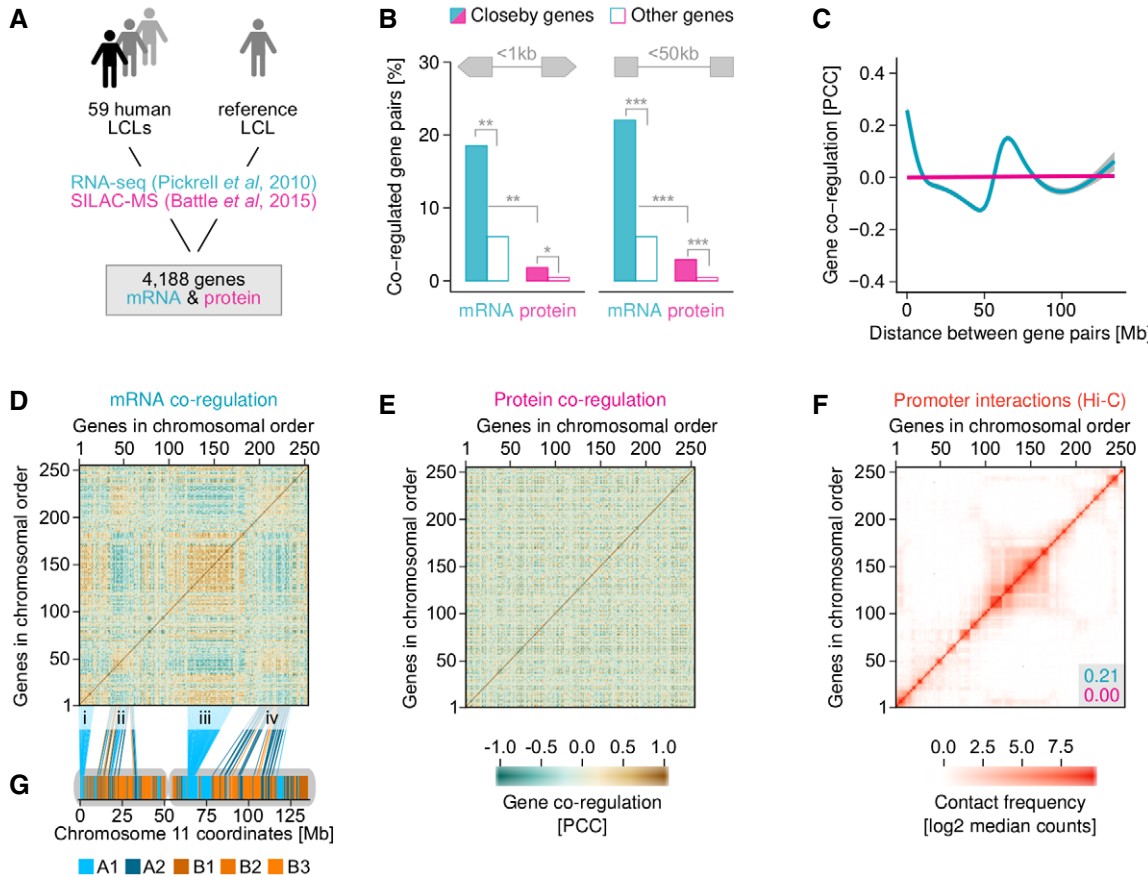

**Figure 1.  Spatial proximity of genes affects mRNA but not protein regulation.**

A   We analysed previously reported mRNA and protein abundances in 59 lymphoblastoid cell lines (LCLs), relative to a reference sample.

B   Genes transcribed from bidirectional promoters frequently have co-regulated mRNA abundances, but only a fraction of these also have co-regulated protein abundances (left). The same is true for non-bidirectional gene pairs whose transcription start sites (TSS) are < 50 kb apart, irrespective of their orientation (right) (*$P < 0.05$, **$P < 2 \times 10^{-7}$, ***$P < 4 \times 10^{-14}$ based on Fisher's exact test).

C   mRNA co-regulation of gene pairs on chromosome 11 decreases with chromosomal distance over many megabases, but not monotonously. Protein co-regulation is unaffected by genomic distance.

D   mRNA co-regulation map for chromosome 11 showing large patches of co-regulated (brown) and anti-regulated (blue) gene pairs. Four large, co-regulated patches are highlighted (i–iv).

E   No regulation patches exist on the protein level.

F   mRNA co-regulation patches partially coincide with physical associations between genes derived from Hi-C data (Rao *et al*, 2014). Numbers in grey box show the Pearson correlation between the Hi-C map and mRNA (blue) or protein (red) co-regulation maps.

G   Patches i, iii and ii, iv broadly coincide with genome subcompartments A1 and A2, respectively.

of the observed cases could be explained through a shared function (Williams & Bowles, 2004).

**Long-range gene co-regulation leads to coordinated mRNA but not protein expression**

The influence of gene distance on co-regulation of transcripts is not limited to genes in close proximity. As seen in the example of chromosome 11, mRNA co-regulation extends over many megabases but does not affect protein abundances (Fig 1C). Although co-regulation generally declines with increasing gene distance, such long-range effects are unlikely to result from transcriptional interference *in cis*. A major co-regulation peak of genes that are more than 50 Mb apart on chromosome 11 suggests that long-range chromosome folding

may be involved. In agreement with this, all chromosomes have distinct co-regulation curves (Appendix Fig S2).

The co-regulation map of chromosome 11 shows large patches of genes whose transcripts are coordinately up- and downregulated (Fig 1D). Importantly, no corresponding co-regulation is observed on the protein level (Fig 1E). However, the mRNA co-regulation map shows a striking similarity to physical associations observed for our gene set, as extracted from existing Hi-C data (Rao *et al*, 2014; Fig 1F). The Hi-C contact matrix of chromosome 11 is correlated with the mRNA co-regulation map (PCC 0.21, $P < 2 \times 10^{-318}$), but not the protein map (PCC 0.00, $P = 0.4$). Similar mRNA co-regulation patches can be observed on other chromosomes (Fig EV3) as well as between different chromosomes (Fig EV4). Generally, both intra- and interchromosomal co-regulation patches

correspond to areas with increased Hi-C contacts (Appendix Table S2). Some chromosomes have more prominent patches than others (Fig EV3). Chromosome 19, which is short but exceptionally gene-dense, is unique in forming a single large co-regulation patch (Fig EV3C). Importantly, none of these mRNA co-regulation patches are reflected at the protein level (Figs EV3 and EV4, Appendix Fig S2). This suggests that regulatory interference between genes that are close in 3D could be associated with similar non-functional mRNA co-regulation as observed for neighbouring genes in the genome sequence.

We next sought to determine which structural features of the genome give rise to mRNA co-regulation patches. Four large mRNA co-regulation patches can be observed on chromosome 11 (labelled i–iv in Fig 1D). Co-regulation patches differ widely in size but often span many megabases, likely reflecting broad architectural features. Notably, promoters and enhancers typically interact on a smaller scale, within topologically associated domains (Gibcus & Dekker, 2013). However, co-regulated groups of genes are more reminiscent of genome compartments. Genome compartments were first identified on the basis of long-range interactions mapped by Hi-C, which showed that open and closed chromatin spatially segregate into two genome-wide compartments (Lieberman-Aiden *et al*, 2009). The compartments containing active and repressive chromatin were designated A and B, respectively. A high-resolution Hi-C map of the genome in LCLs subsequently identified that these compartments segregate further into six subcompartments: A1-2 and B1-4 (Rao *et al*, 2014). Genomic loci within each subcompartment tend to be associated with each other more often than with loci from other subcompartments, that is they are in closer spatial proximity. We find that co-regulation patches i and iii of chromosome 11 align with subcompartment A1 and patches ii and iv align with subcompartment A2 (Fig 1G). These are the two subcompartments of the genome formed by transcriptionally active chromatin, which is expected given that we analyse housekeeping genes. Interestingly, genes across patches i and iii are co-regulated, as are genes across patches ii and iv, suggesting that co-localisation in subcompartments may contribute to the existence of these patches.

### Genes with co-regulated mRNAs co-localise in genome subcompartments

To assess systematically the overlap of co-regulated gene groups with genome compartments, we clustered genes by co-regulation. We found four transcriptome regulation groups T1-4 (Fig 2A and Dataset EV3), explaining more than 50% of the total variance (Appendix Fig S3). Transcripts within each group are co-regulated (Fig 2A and B). Genes from T1 and T2 are strongly enriched for subcompartments A2 and A1, respectively (Fig 2C). Curiously, they are anti-correlated, that is when T1 genes are upregulated, T2 genes tend to be downregulated, and vice versa (Fig 2B). Co-regulated genes of the T3 and T4 groups are also enriched for A1 and A2 subcompartments, respectively. However, they are independent of T1 and T2, that is there is neither a positive nor a negative correlation between T1/T2 and T3/T4 (Fig 2B). Therefore, while subcompartments A1 and A2 are strongly related to transcriptome regulation groups, they are not sufficient to explain them.

Genome compartments and subcompartments were defined solely based on their physical interaction patterns, but also have

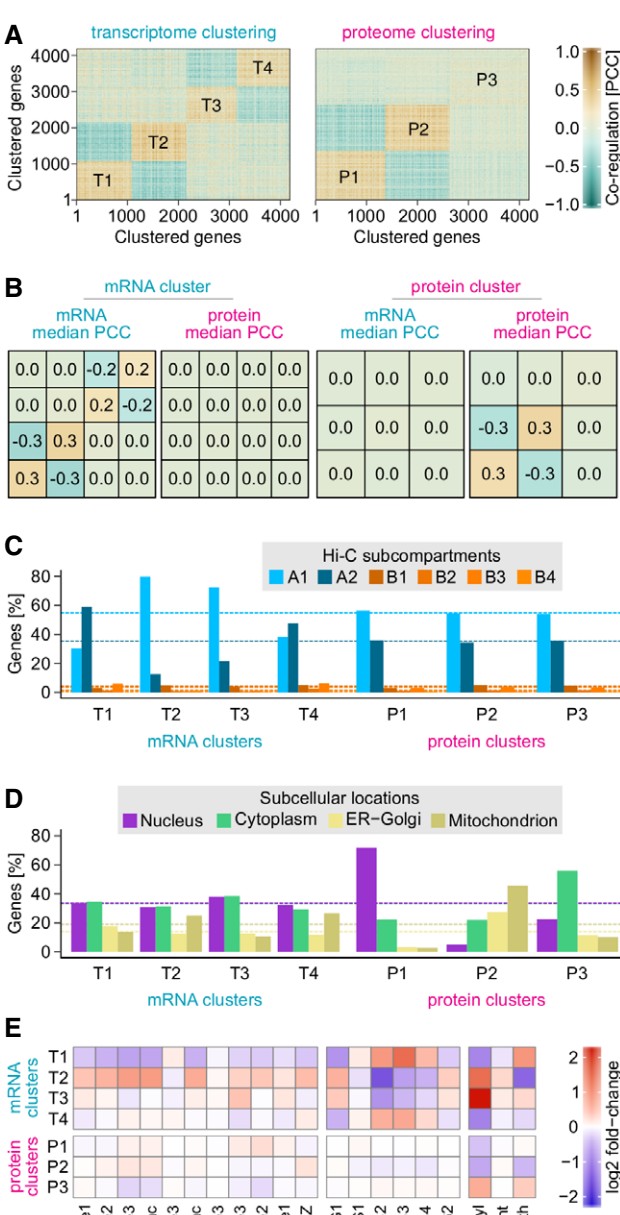

**Figure 2. Transcriptome and proteome regulation are driven by different factors.**

A   *k*-means clustering of genes based on their mRNA or protein abundance changes across LCLs.

B   Median Pearson's correlation coefficients (PCCs) for each transcriptome and proteome *k*-means cluster. Genes assigned to different *k*-means clusters can either be anti-regulated (e.g. T1 and T2) or not correlated (e.g. T1 and T3). *k*-means clusters formed by genes that are co-regulated at the mRNA level are not generally co-regulated at the protein level, and vice versa.

C   Transcriptome clusters are strongly enriched for subcompartment A1 or A2. Dashed lines indicate the percentage of genes expected if subcompartments were evenly distributed across clusters.

D   Proteome clusters are mainly composed of proteins from distinct subcellular locations. Dashed lines indicate the percentage of genes expected if subcellular locations were evenly distributed across clusters.

E   Genomic and epigenomic features enriched in each cluster relative to the whole dataset.

different genomic and epigenomic characteristics. A1 and A2 subcompartments are both enriched for features associated with transcriptionally active chromatin, but to different extents (Rao *et al*, 2014). Interestingly, we also found clear differences in histone modifications and DNA methylation associated with transcriptome regulation groups (Fig 2E). For example, in comparison with T2, T1 gene bodies are enriched for H3K9me3, depleted in activating marks such as H3K4me3 and H3K27ac, are longer, replicate later and have a lower GC content. These differences mirror those observed between A2 and A1 subcompartments (Rao *et al*, 2014). In contrast, T3 and T4 do not show these features despite preferentially localising to A1 and A2 subcompartments. Instead, T3 genes display heavy CpG methylation, which is almost an order of magnitude stronger than for T4 genes. Consequently, T3 and T4 define their own epigenetic subpopulation within A-type compartments.

## Genes with co-regulated protein abundances are related functionally, not spatially

Clustering analysis of protein expression profiles led to three proteome regulation groups P1-3 (Fig 2A and Dataset EV3), explaining more than 50% of the total variance (Appendix Fig S3). Neither genome compartments nor epigenomic signatures appear to be associated with proteome regulation groups (Fig 2C and E). In contrast, proteome regulation groups broadly correspond to subcellular locations: nucleus (P1), mitochondria, ER and Golgi (P2) and cytoplasm (P3) (Fig 2D). They are also enriched for biological processes taking place in these subcellular locations (Appendix Fig S4). In contrast, T1-4 only weakly coincide with subcellular locations or biological processes.

Intriguingly, T1-4 and P1-3 are independent of each other, that is genes that are clustered based on their transcript expression signature are generally not co-regulated on the protein level, and vice versa (Fig 2B). This suggests that much of the mRNA coexpression of genes from the same subcompartment may be non-functional. Note that as for sequence proximity (see above), this appears to contrast with a previous report that genes which are close in 3D nuclear space often have similar functions (Thévenin *et al*, 2014). However, we also find significant enrichment of functional associations between genes from the same subcompartment (Fig EV2B). Nevertheless, in quantitative terms, the extent of mRNA co-regulation strongly exceeds co-function as well as protein co-regulation. For example, while 11% of gene pairs in the same (intrachromosomal) subcompartment have co-regulated mRNAs, < 1% have similar functions according to STRING and are co-regulated at the protein level (Fig EV2B).

## Gene clustering within but not between chromosomes associates with reduced expression noise

In yeast, clustering of genes in the genome sequence is associated with reduced expression noise (Batada & Hurst, 2007; Wang *et al*, 2011). However, the situation is more complex when considering the 3D structure of the genome. Highly transcribed gene clusters tend to form fewer contacts with other chromosomes, and genomic loci with more interchromosomal contacts tend to have higher expression noise (McCullagh *et al*, 2010; Sandhu, 2012).

We tested whether gene clustering has a similar effect in human cells. For each gene in our dataset, we calculated a clustering degree, defined as the average distance to its three nearest neighbouring genes along the DNA sequence. We then compared the expression noise of the 5% most and least clustered genes, respectively. As observed in yeast, we find that gene expression noise in LCLs is significantly reduced for genes in gene-dense areas (Fig 3A). The noise-reducing effect is much more significant on the mRNA than the protein level.

In a second step, we investigated whether gene clustering in nuclear space has a similar noise-reducing effect. In principle, gene-dense regions may interact with each other in 3D to benefit from further noise reduction by forming "super-clusters". The three human histone gene clusters on chromosome 6, for example, converge in 3D to form such a super-cluster (Sandhu *et al*, 2012). Therefore, we calculated a second clustering degree for each gene, defined as the average distance to its three nearest neighbours in 3D, using Hi-C contacts. To capture long-range interactions resulting from chromosome folding, we only considered neighbouring genes that were on the same chromosome, but at least 500 kb up- or downstream in terms of DNA sequence. There is a positive correlation between the clustering degree in 1D and 3D (PCC 0.32, $P < 6 \times 10^{-97}$), suggesting that genes clustered along the sequence are also more densely packed in the 3D structure of a chromosome. Moreover, this gene clustering due to chromosome folding is also associated with a significant reduction of gene expression noise, albeit not as strongly as sequence-based clusters (Fig 3A).

Next, we investigated clusters that genes from different chromosomes may form in nuclear space, calculating a third clustering degree based on interchromosomal Hi-C contacts. As shown in yeast (McCullagh *et al*, 2010; Sandhu, 2012), we find a negative correlation between sequence-based and interchromosomal clustering (PCC $-0.1$, $P < 5 \times 10^{-11}$). This suggests that gene-dense regions, while forming long-range, noise-reducing interactions within the same chromosome, are less likely to interact with gene clusters on a different chromosome. Moreover, genes forming interchromosomal clusters are associated with higher expression noise than those with fewer interactions (Fig 3A). This difference is not statistically significant but is in agreement with earlier findings in yeast (McCullagh *et al*, 2010; Sandhu, 2012).

## Coexpression of closeby genes is driven by stochastic epigenetic fluctuations and regulatory interference

How can gene proximity lead to mRNA coexpression? Many incidents of coexpressed genes that are close in sequence have been linked to stochastic alternation between an active and inactive chromatin state (Batada *et al*, 2007). Such chromatin fluctuations can lead to coordinated transcriptional bursts of all genes within a chromatin domain (Raj *et al*, 2006). We first compared the chromatin environment of genes that are co-regulated with their sequence neighbours with genes that show no such co-regulation ("neighbours" being defined as genes whose TSSs are < 50 kb away). We find that genes which are coexpressed with their neighbours are more often flanked by heterochromatin, upstream of their transcription start site (Fig 3B). This is consistent with mRNA coexpression driven by stochastic spreading of the adjacent heterochromatin domain into the active locus, silencing all genes therein. This is

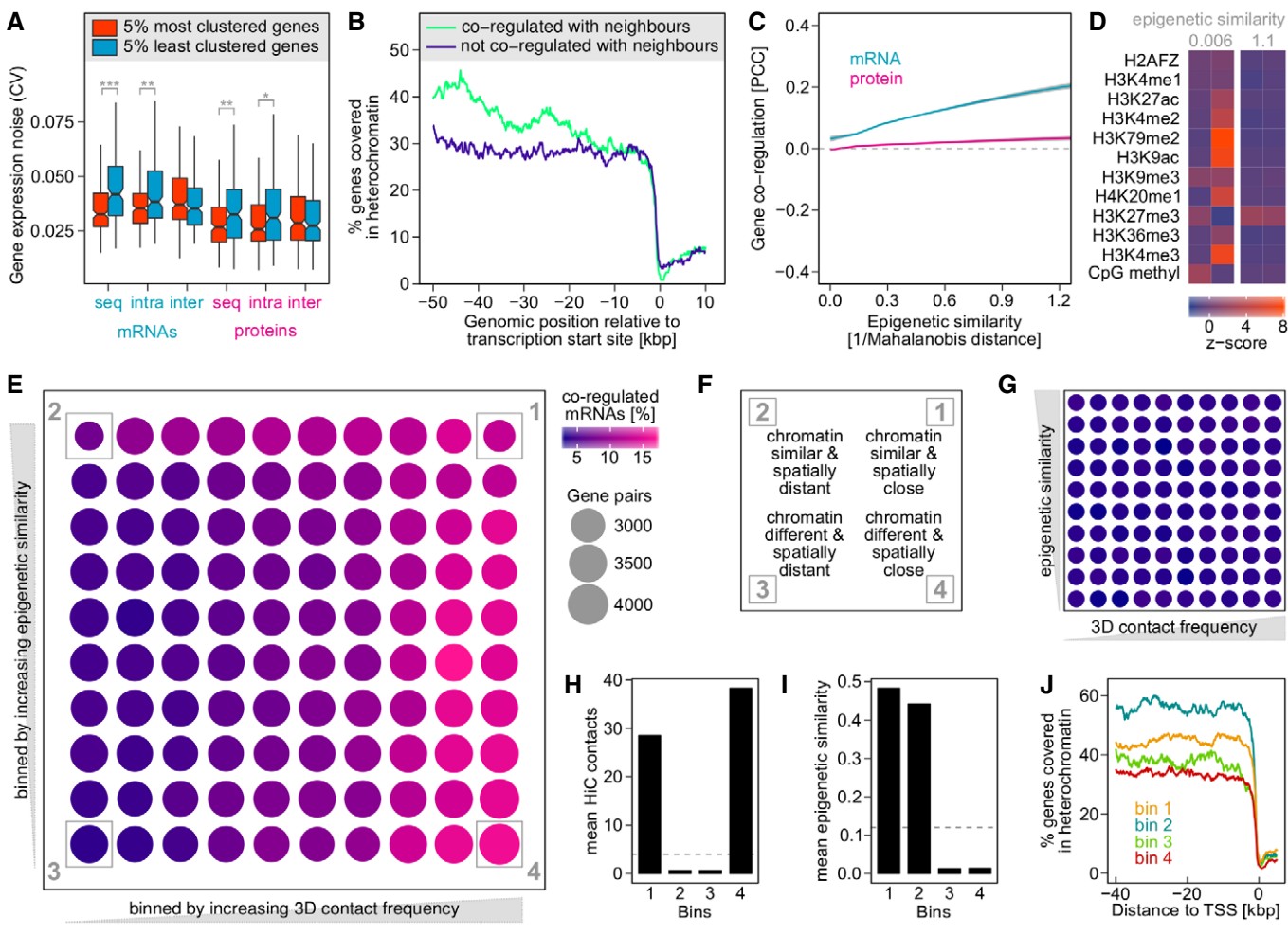

**Figure 3.   mRNA coexpression of neighbouring genes is driven by chromatin fluctuations and regulatory interference.**

A   Intrachromosomal gene clustering reduces gene expression noise. We determined the expression noise (coefficient of variation, CV) of the most and least densely clustered genes, considering three different types of clustering: in terms of sequence proximity (seq), using long-range Hi-C contacts (> 500 kb) within the same chromosome (intra) and using interchromosomal Hi-C contacts (inter). Expression noise is reduced for clustered genes, except for genes forming more interchromosomal contacts (*P < 0.01, **P < 0.002, ***P < 5 × 10$^{-6}$ based on Kolmogorov–Smirnov test). Boxplot drawn in the style of Tukey, that is box limits indicate the first and third quartiles, central lines the median, whiskers extend 1.5 times the interquartile range from the box limits. Notches indicate the 95% confidence interval for comparing medians.

B   The upstream region of genes that are co-regulated with their neighbours, that is other genes within 50 kb, is more likely to be occupied by heterochromatin than that of genes showing no such co-regulation. Heterochromatin regions in LCLs have been reported previously (Ernst *et al*, 2011).

C   Epigenetic similarity calculated on the basis of histone marks and CpG methylation is a strong general predictor of mRNA co-regulation. Curves are fitted to all intrachromosomal gene pairs irrespective of their genomic distance.

D   Two randomly picked gene pairs exemplifying low and high epigenetic similarity, respectively. Each column represents a gene and each row an epigenetic feature. Colours show the standardised, average abundance of each mark across the gene body.

E   mRNA co-regulation requires epigenetic similarity or spatial proximity, but not both. Intrachromosomal gene pairs were binned by epigenetic similarity and spatial proximity (Hi-C contacts), and the percentage of co-regulated mRNAs is shown in colour. Note bins 2 and 4 are both enriched for co-regulated mRNAs despite containing gene pairs that are spatially distant and epigenetically different, respectively.

F   Description of bins highlighted in panel (E).

G   Gene pairs binned as in (E) but colour showing percentage of co-regulated proteins. Protein co-regulation does not depend on epigenetic similarity or spatial proximity.

H   On average, gene pairs in bins 1 and 4 have many more Hi-C contacts than those in bins 2 and 3, that is they are spatially closer. Dashed line shows average Hi-C contacts between genes in the dataset.

I   On average, gene pairs in bins 1 and 2 are epigenetically much more similar than those in bins 3 and 4. Dashed line shows average epigenetic similarity between genes in the dataset.

J   Heterochromatin profile for genes in bins 1–4.

reminiscent of subtelomeric regions in yeast, which are hot spots for expression noise (Batada & Hurst, 2007) due to transient spreading of telomeric heterochromatin (Anderson *et al*, 2014).

Notably, chromatin fluctuations may lead to mRNA coexpression that is not restricted to genes in close spatial proximity. Chromatin factors play a key role in creating gene expression noise (Newman

*et al*, 2006). Fluctuating expression levels of, for example, a histone-modifying enzyme may simultaneously affect all its target chromatin domains in the genome. To test for such a global chromatin-mediated co-regulation effect, we determined the epigenetic similarity between all genes in our dataset. We defined "epigenetic similarity" based on the abundance of various histone marks within gene bodies. We used the Mahalanobis distance to measure similarity, as this takes into account that some histone marks are strongly co-dependent, for example H3K9ac and H3K4me3. Genes with similar epigenetic profiles are targeted by a similar set of chromatin-modifying factors, and are therefore expected to respond similarly to stochastic fluctuations of these factors. Indeed, we find that the epigenetic similarity is a strong predictor of non-functional mRNA co-regulation (Fig 3C and D).

This chromatin fluctuation scenario is a passive mechanism where genes simply respond to changes in their chromatin domain. However, on a local scale, transcriptional changes of one gene may directly affect the transcription of its neighbours, if chromatin remodelling or transcription factors spill over to adjacent genomic regions (Ebisuya *et al*, 2008; Wang *et al*, 2011). This "regulatory interference" model crucially depends on spatial proximity, but does not require co-regulated genes to be part of the same chromatin domain. To compare the impact of chromatin and gene distance on non-functional mRNA coexpression, we grouped gene pairs based on epigenetic similarity as well as based on Hi-C contact frequency. We then observed which groups contain co-regulated mRNAs (Fig 3E). This shows that gene pairs which are far apart both spatially and epigenetically are rarely co-regulated (bin 3 in Fig 3E and F). Gene pairs with similar histone marks tend to be co-regulated, even if they are spatially distant (Fig 3E and H). Co-regulation of such genes is consistent with the passive chromatin fluctuation model, but not the transcriptional interference model. Importantly, spatially close gene pairs can be co-regulated even if their histone marks show no similarity (bin 4 in Fig 3E and I). This type of coexpression is not consistent with the passive chromatin fluctuation model, since the epigenetic differences between the gene pairs suggest that, in steady state, they occupy distinct chromatin domains. These genes are also the least likely to be flanked by heterochromatin (Fig 3J). However, the behaviour of gene pairs in bin 4 is consistent with the regulatory interference model, where fluctuations in one gene affect the chromatin and transcriptional state of its neighbours, in sequence and 3D. Note that this effect is buffered at the protein level (Fig 3G), which is in agreement with this type of coexpression being not functional.

### Buffering of non-functional mRNA coexpression tends to be a non-selective process

Finally, we asked which post-transcriptional mechanisms might buffer the coexpression of genes that are spatially close, but functionally unrelated. In principle, this could be a selective process that specifically targets closeby genes and disentangles their expression patterns. Alternatively, buffering could be a neutral process, where the lack of coordination between post-transcriptional mechanisms prevents the mRNA coexpression to be propagated to the protein level. In this case, a selective process would need to exist to ensure that functionally related genes do in fact have co-regulated protein abundances. To distinguish between these two possibilities, we analysed five measures of post-transcriptional gene expression control (Fig 4).

First, we tested whether gene pairs with sustained protein co-regulation are more likely to have similar mRNA half-lives in LCLs (Duan *et al*, 2013), relative to co-regulated gene pairs with buffered protein abundances. Indeed, we find this to be the case, even though the difference is modest (Fig 4A). Next, we analysed which co-regulated gene pairs are more likely to be targeted by the same miRNA (Helwak *et al*, 2013). Again, gene pairs that are also co-regulated on the protein level are enriched for pairs sharing at least one miRNA. Third, as an indication for translation-related effects, we took into account ribosome profiling data for the LCL cell line panel (Battle *et al*, 2015), which reflect both the abundance of mRNAs and the extent to which they are occupied by ribosomes (Ingolia, 2014). Gene pairs with coexpressed proteins are almost three times as likely to have correlated ribosome profiles than pairs which only have co-regulated mRNA abundances. Then, we looked at the impact of protein degradation, by considering the occurrence of non-exponentially degraded proteins (NEDs) (McShane *et al*, 2016). These are proteins that are rapidly degraded after synthesis, for example because they are protein complex subunits produced in super-stoichiometric amounts. Again, we find that NEDs are enriched among gene pairs with co-regulated proteins rather than those with buffered protein levels. Finally, we show that the protein sequence length, which strongly correlates with the extent of post-transcriptional control (Vogel *et al*, 2010), is more similar for co-regulated than buffered proteins. Proximity in the genome seemed to have no impact on the similarity of gene pairs in any of the five measures of post-transcriptional gene expression control investigated here (Fig 4B). Taken together, these results suggest that buffering of co-regulated closeby genes may occur via a neutral mechanism, with buffered gene pairs consistently lacking the extent of shared post-transcriptional processing observed for functionally related gene pairs. If mRNA coexpression is functionally relevant, multiple layers of post-transcriptional control appear to work together to ensure that this is propagated to the protein level.

## Discussion

Genes are not randomly distributed across the sequence and structure of the genome, forming clusters that tend to be coexpressed but do not generally have a shared function. Gene expression noise is detrimental to cell fitness, especially for housekeeping genes (Fraser *et al*, 2004). Clusters of actively transcribed genes have low expression noise, which may drive the evolution of non-random gene order (Batada & Hurst, 2007). The coexpression of functionally unrelated neighbouring genes may then be a side effect of the selection for noise reduction. However, such coexpression is not necessarily deleterious. As we show here, non-functional co-regulation is frequently observed at the mRNA level, but is largely buffered at the protein level. Consequently, non-functional coexpression is unlikely to offset the benefit of noise reduction.

The expression profiles of genes in a cluster co-evolve, such that the evolutionary change in expression of one gene on average predicts changes in its neighbours (Ghanbarian & Hurst, 2015).

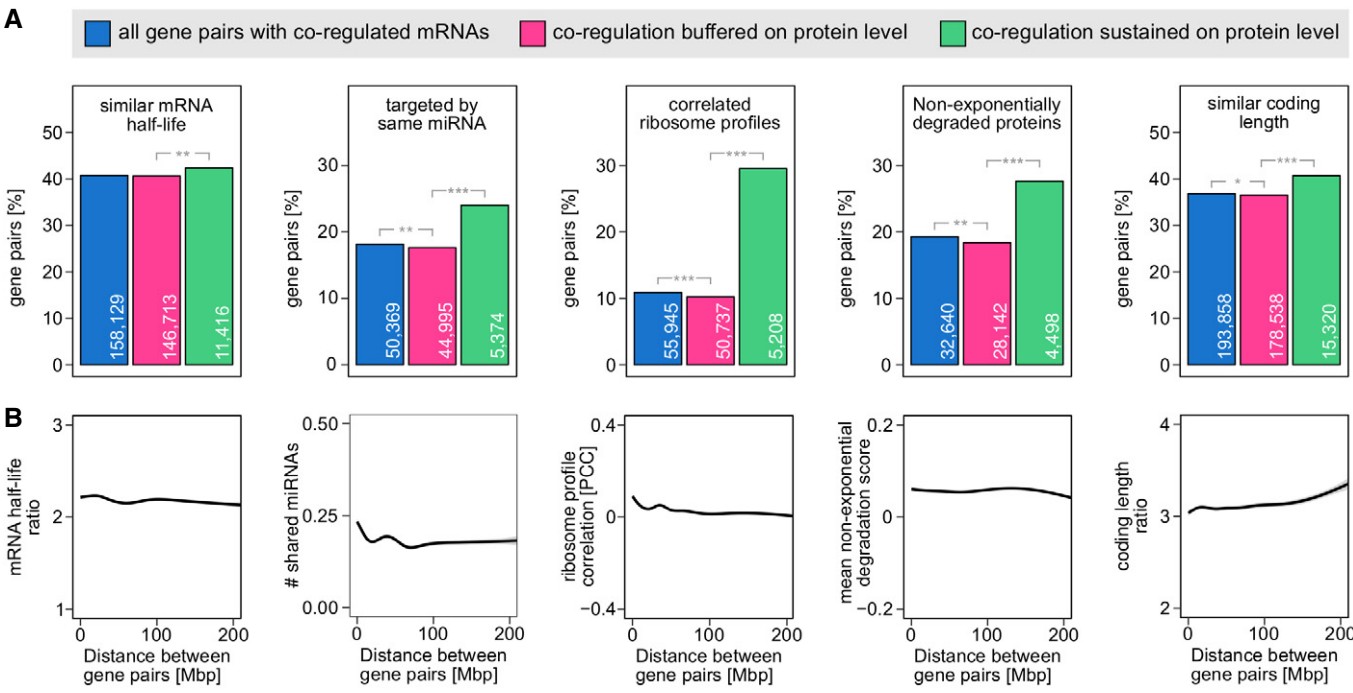

**Figure 4.   Buffering of non-functional mRNA co-regulation likely is a passive process.**

A  Percentage of gene pairs with coordinated post-transcriptional regulation, irrespective of genomic distance. Gene pairs with sustained protein co-regulation consistently stand out as more likely to share similar aspects of post-transcriptional control. Genes were considered to have a similar mRNA half-life if the half-life ratio between the more and less stable gene was < 1.5. For miRNAs, all gene pairs targeted by at least one shared miRNA were considered. Gene pairs were said to have correlated ribosome profiles if their ribosome occupancy correlated with PCC > 0.5 (BH adj. *P* < 0.001) across LCLs. For the non-exponentially degraded proteins (NEDs) barchart, gene pairs containing at least one NED were counted. Coding length was considered similar if the longer protein was < 1.5-fold longer than the shorter protein. Numbers of gene pairs are shown inside the bars. Statistical significance was calculated using Fisher's exact test (*$P$ < 0.01, **$P$ < 1 × 10$^{-6}$, ***$P$ < 3 × 10$^{-27}$).

B  No striking relationship between gene distance and the extent to which gene pairs show similar post-transcriptional regulation. Note that the small increase of similar ribosome occupancy towards closeby genes may be explained by the fact that ribosome profiles partially reflect mRNA abundance.

Nevertheless, it is still unclear whether expression clusters are the result of natural selection. In yeast, only the most highly coexpressed neighbours are conserved as a pair, but these also tend to be functionally related (Hurst *et al*, 2002). Neighbouring gene pairs that separate tend to show interchromosomal co-localisation (Dai *et al*, 2014). In *Drosophila*, highly coexpressed neighbouring gene pairs are less likely to be conserved than expected (Weber & Hurst, 2011). In mammals, although some coexpression clusters are evolutionarily maintained (Sémon & Duret, 2006), natural selection generally tends to separate gene pairs that show a strong position-related coexpression effect (Liao & Zhang, 2008) or that involve tissue-specific expression (Lercher *et al*, 2002). This indicates that non-functional coexpression can affect cell fitness under some circumstances, possibly if it becomes so strong that it persists through the uncoordinated post-transcriptional processes.

The existence of coexpression clusters may also reflect the way new genes originate. For example, highly transcribed chromatin regions are more susceptible to retroposition (Hurst *et al*, 2004). Recently, it has been proposed that the large number of human gene pairs in head-to-head orientation may arise from divergent transcription of single genes, when initially noncoding, antisense transcripts evolve into new protein-coding genes (Wu & Sharp, 2013). In both of these cases, new genes would have no sequence homology with their neighbours, and would therefore be unlikely to share

their function. However, some of the most well-known coexpression clusters, such as histone gene clusters, arose by gene duplication. Gene duplicates could potentially explain why some gene clusters are functionally related. There are 30 gene pairs in our dataset that are located within 50 kb from each other and are coexpressed on both the mRNA and the protein level. Of these, 10 (33%) are classified as paralogues by Ensembl, a strong enrichment considering that paralogues account for only 1.5% of these closeby gene pairs overall. However, 20 (66%) of the clustered gene pairs with co-regulated protein abundances show no evidence for paralogy, suggesting that functionally relevant clusters need not necessarily arise by gene duplication.

Our analysis focussed on housekeeping genes, because comparable data for tissue- or condition-specific genes were not available. Housekeeping genes constitute about half of all human genes (Uhlén *et al*, 2015). They have a higher tendency to cluster than other genes (Lercher *et al*, 2002), presumably because they are more sensitive to gene expression noise (Fraser *et al*, 2004). Interestingly, post-transcriptional expression control is particularly important for housekeeping genes (Gandhi *et al*, 2011; Jovanovic *et al*, 2015). Notably, transcriptional activation of induced genes can also lead to co-activation of functionally unrelated neighbouring genes (Spitz *et al*, 2003; Ebisuya *et al*, 2008). However, it remains to be seen if such co-activation is also buffered at the protein level.

In conclusion, non-functional mRNA coexpression, due to chromatin fluctuations and regulatory interference, is far more common than previously thought. Generally, this does not hamper cell fitness as post-transcriptional regulatory mechanisms enforce functional coexpression while dampening non-functional coexpression. Our observations suggest that evolution of human genome organisation is driven by noise reduction, which is a hypothesis initially made in yeast (Batada & Hurst, 2007). The large presence of non-functional coexpression of genes at the transcript but not protein level has implications for the fields of transcriptomics and proteomics when screening for functional links between genes.

# Materials and Methods

### mRNA abundances in human lymphoblastoid cell lines

RNA-sequencing data for human lymphoblastoid cell lines (LCLs) have been reported (Pickrell *et al*, 2010). Counts per mapped reads were downloaded from http://eqtl.uchicago.edu and converted to log2 "reads per kilobase transcript per million mapped reads" (RPKMs). Genes expressed in < 30 LCLs were removed. In order to make mRNA measurements comparable to proteomics data, expression levels needed to be analysed relative to the same reference LCL. To do so, log2 RPKMs values from the reference cell line GM19238 were subtracted from all other LCLs.

### Protein abundances in human lymphoblastoid cell lines

Protein abundances in LCLs have also been reported (Battle *et al*, 2015). They have been measured by mass spectrometry and quantified relative to the reference cell line GM19238, using stable isotope labelling by amino acids in cell culture (SILAC) (Ong *et al*, 2002). Mass spectrometry raw files were downloaded from the PRIDE repository (Vizcaíno *et al*, 2016) (project identifier PXD001406) and re-processed using MaxQuant 1.5.2.8 (Cox & Mann, 2008). Raw files tagged as "run2" were omitted. Mass spectra were searched against human Swiss-Prot sequences downloaded from Uniprot (UniProt Consortium, 2015). To facilitate combining mRNA and protein datasets, no protein isoforms were considered. We used non-normalised SILAC ratios obtained by MaxQuant with at least two ratio counts. Because the internal standard had been used as heavy SILAC sample, heavy/light (H/L) SILAC ratios were inverted to obtain L/H ratios (i.e. test LCLs / reference LCL). Proteins that could not be unambiguously mapped to a single gene were removed, as were proteins detected in 30 LCLs or less. SILAC ratios were also log2-transformed.

### Combining mRNA and protein expression data

To combine mRNA and protein data, ENSEMBL gene IDs from RNA sequencing were mapped to Uniprot IDs using Uniprot's webtool (UniProt Consortium, 2015). Genes with ambiguous mappings were removed. We also only considered LCLs for which both mRNA and protein data were available. The resulting file contains mRNA and protein abundances for 4,188 human genes in 59 LCLs, relative to the GM19238 reference sample (Dataset EV1). It contains 0.1 and 6.7% missing values for mRNA and protein measurements, respectively.

### Defining positions of genes in the genome

Genomic coordinates of human genes (dataset version GRCh38.p5) were downloaded from ENSEMBL (Yates *et al*, 2016). As we are considering genes but not specific transcript or protein isoforms, transcription start sites (TSSs) were defined as the start site of the outermost transcript of a gene.

### Testing gene pairs for co-regulation

Coordinated up- and downregulation of gene expression was measured using Pearson's correlation coefficient (PCC). The gene expression datasets for LCLs (Dataset EV1) were used as input. The median log2 fold change of each LCL was set to zero, in order to prevent correlations reflecting irrelevant data features such as uneven mixing of light and heavy SILAC samples. Gene pairs were considered to be co-regulated at PCC > 0.5, but only if the correlation was significant (Benjamini and Hochberg-adjusted *P*-values < 0.001).

### Characterisation of genes as housekeeping genes

To demonstrate that the 4,188 genes in the LCL dataset belong to the constitutively expressed core proteome, we performed a number of tests:

#### Chromatin states of gene promoters
Chromatin states of the genome of the GM12878 lymphoblastoid cell line were determined previously (Ernst *et al*, 2011). They were downloaded as hg19 genome coordinates from the USCS genome browser (Rosenbloom *et al*, 2015) and converted to GRCh38 coordinates using the liftOver command line tool (available at https://genome-store.ucsc.edu/). Genomic regions with conflicting chromatin state annotations, resulting from the genome coordinates update, were removed. For each gene in our dataset, the chromatin state mapping to its transcription start site was determined.

#### GO term enrichment
A statistical overrepresentation test was performed using the PANTHER classification system (Mi *et al*, 2016) according to the reported protocol (Mi *et al*, 2013). Overrepresentation of Gene Ontology Biological Process (slim) terms was assessed for our 4,188 genes compared to the entire human genome. Only significantly enriched terms (more than twofold; *P* < 0.05 after Bonferroni correction) were considered.

#### mRNA tissue expression data
mRNA expression levels in different human tissues have been assessed using RNA sequencing (Uhlén *et al*, 2015). Transcripts detected with FPKM ≥ 1 were considered to be expressed.

#### Protein tissue expression data
Protein expression levels in different human tissues have been assessed using mass spectrometry (Wilhelm *et al*, 2014) (available at www.proteomicsdb.org). To avoid bias due to the incomplete nature of current proteome maps, only tissues with expression values for more than 6,000 proteins were considered.

## Defining pairs of genes with related functions (focussed on accuracy)

To test whether genes with related functions are co-regulated across LCLs, we defined three sets of functionally linked gene pairs. Functional associations in these test sets are as accurate—not as comprehensive—as possible.

### Gene pairs from same protein complexes

Human protein–protein interaction pairs based on Reactome pathways (Fabregat *et al*, 2016) were downloaded from www.reactome.org (homo_sapiens.interactions.txt file; March 2016). They were filtered for physical interactions of the "direct_complex" category. Gene pairs belonging to more than one complex and homodimeric interactions were removed.

### Gene pairs encoding enzymes from consecutive metabolic reactions

As for protein complexes, human protein–protein interaction pairs based on Reactome pathways (Fabregat *et al*, 2016) were downloaded from www.reactome.org (homo_sapiens.interactions.txt file; March 2016). They were filtered for interactions of the "neighbouring_reactions" category. These are interactions where one gene/protein produces the input or catalyst for the second reaction. Any gene pairs known to interact also physically, that is belonging to the "direct_complex" or "indirect_complex" categories, were removed. In addition, gene pairs were filtered for those involved in *metabolic* pathways, as opposed to, for example, the cell cycle pathway which would contain irrelevant reactions such as "Mis18 complex binds the centromere". To do so, we first inferred all pathways mapping to the metabolism root pathway, using the pathway hierarchy relationship file (ReactomePathwaysRelation.txt, available on www.reactome.org). Enzymatic reactions belonging to each metabolic pathway were then identified using another interaction file available from Reactome (homo_sapiens.mitab.interactions.txt). Finally, to avoid "trivial" consecutive reactions such as those involving ubiquitous metabolites like $NAD^+$, we removed metabolic reactions with more than ten neighbouring reactions.

### Gene pairs from identical subcellular locations

Subcellular localisations of human proteins were downloaded from Uniprot (UniProt Consortium, 2015). Proteins localising to more than one subcellular location were removed. To avoid trivial localisations such as "cytoplasm", only subcellular compartments with 200 or less known protein components were considered.

## Defining pairs of genes with related functions (focussed on completeness)

To estimate an upper limit for how many coexpressed neighbouring genes may be functionally related, we defined a separate test set based on the STRING database (Szklarczyk *et al*, 2017). Functional associations in this test set are as comprehensive as possible. Protein network data for *Homo sapiens* were downloaded from http://string-db.org. We considered all functional associations with a combined STRING score > 0.7. This score integrates various types of evidence and indicates the likelihood of the association to be biologically meaningful, specific and reproducible.

## Testing functionally related gene pairs for co-regulation

Correlation coefficients were obtained for every gene pair in our three test sets (protein complexes, consecutive metabolic reactions, subcellular locations) and their distribution was displayed in histograms. As a control, gene pairs were randomly shuffled to break the link between the pairs. For example, gene pairs encoding subunits of the same protein complexes were shuffled such that the same genes were paired randomly, in which case most gene pairs encode subunits of different protein complexes. The Kolmogorov–Smirnov test was used to assess whether PCC distributions of relevant gene pairs were significantly different from those obtained with randomised pairs.

## Chromosome co-regulation mapping

PCCs were calculated for all relevant gene combinations, as described for histograms above. For chromosome co-regulation curves, PCCs were plotted against the genomic distance between transcription start sites, with curves fitted by a generalised additive model. For chromosome co-regulation maps, genes were plotted in their chromosomal order and PCCs between all gene combinations were represented by a colour scale.

## Hi-C interactions for our gene set

Hi-C contact matrices for a lymphoblastoid cell line (Rao *et al*, 2014) were downloaded from NCBI GEO database (accession GSE63525). An unpublished script from Liz Ing-Simmons (available at https://github.com/liz-is/readhic) was adapted (available at https://github.com/Rappsilber-Laboratory/readhic) and then used to import the Hi-C contact matrices into R, using 10-kb resolution and "KRnorm" normalisation for intrachromosomal pairs and 50-kb resolution and "INTERKRnorm" normalisation for interchromosomal pairs. All reads used passed the MAPQ>0 filter. Hi-C data are based on GRCh37 genome coordinates. GRCh37 transcription start sites for all genes were obtained using the biomaRt R package (Durinck *et al*, 2009), considering only the TSS of the outermost transcript of each gene. The GenomicInteractions R package (Harmston *et al*, 2015) was used to determine the contact frequency between the genes in our dataset, considering the median read count of all Hi-C pixels in a range ± 40 kb around the TSS of each gene.

## Analysis of genome subcompartments

Nuclear subcompartments A1, A2, B1, B2, B3 and B4 have been defined previously (Rao *et al*, 2014). A genome-wide mapping of subcompartments in a lymphoblastoid cell line is available via the NCBI GEO database (accession GSE63525). Subcompartment annotations were lifted from hg19/b37 to GRCh38 genome coordinates using the UCSC genome browser service (Rosenbloom *et al*, 2015).

## *k*-means clustering of transcript and protein expression changes

*k*-means clustering was performed using the default algorithm and settings in R (R Core Team, 2016), with $k = 4$ (mRNAs) or $k = 3$ (proteins) and five random start sets. Values of $k$ were chosen such that the clusters explain at least 50% of the total variance.

## Analysis of cluster features

### Subcellular locations

To get a broad understanding of subcellular locations enriched in *k*-means clusters, we downloaded all Uniprot entries mapping to the locations Nucleus (Uniprot subcellular location ID: SL-0191), Endoplasmic reticulum (SL-0095), Golgi apparatus (SL-0132), Mitochondrion (SL-0173) and Cytoplasm (SL-0086) (UniProt Consortium, 2015). Proteins localising to the Endoplasmic reticulum and/or the Golgi apparatus were combined as "ER-Golgi". Proteins mapping to more than one organelle were removed.

### GO term enrichment

A statistical overrepresentation test was performed using the PANTHER classification system (Mi *et al*, 2016) according to the reported protocol (Mi *et al*, 2013). Overrepresentation of Gene Ontology Biological Process (complete) terms in each cluster, relative to other clusters, was assessed. Using PANTHER's GO hierarchy annotation, we reported only the most specific GO terms and omitted any co-enriched parent terms for clarity. All reported GO terms were significantly enriched ($P < 0.05$ after Bonferroni correction).

### Genomic and epigenomic features

Raw signals of ChIP-seq experiments for lymphoblastoid cells were downloaded from ENCODE (ENCODE Project Consortium, 2012) in hg19 genomic coordinates. ENCODE accessions were ENCFF000ARW (H2AZ), ENCFF000ARZ (H3K4me1), ENCFF000ATL (H3K4me2), ENCFF001EXX (H3K4me3), ENCFF000ASJ (H3K27ac), ENCFF000ATX (H3K79me2), ENCFF000AUF (H3K9ac), ENCFF000AUL (H3K9me3), ENCFF000AUS (H4K20me1), ENCFF001EXC (H3K27me3), ENCFF001EXP (H3K36me3), ENCFF001GNK (RepliSeq G1b), ENCFF001GNN (RepliSeq G2), ENCFF001GNR (RepliSeq S1), ENCFF001GNT (RepliSeq S2), ENCFF001GNX (RepliSeq S3) and ENCFF001GOA (RepliSeq S4). These bigWig files were converted to bedGraph files, lifted over to GRCh38 coordinates, cleared of any resulting overlaps and converted back to bigWig files using command line tools from the UCSC genome browser (Rosenbloom *et al*, 2015) (tools available at https://genome-store.ucsc.edu/). GC percentage over 5-bp windows was downloaded from the UCSC genome browser (Rosenbloom *et al*, 2015). Average signals over gene bodies were calculated with the UCSC bigWigAverageOverBed command line utility, using the coordinates of our genes as bed files. CpG methylation from reduced representation bisulphite sequencing of a lymphoblastoid cell line was also available from ENCODE (ENCODE Project Consortium, 2012) (experiment ENCSR000DFT; file accession ENCFF001TLQ). After lifting the hg19 bedMethyl file over to GRCh38 genomic coordinates, the mean percentage of CpG methylation in gene bodies was calculated using an R script. For each epigenomic or genomic feature, the median enrichment for genes in each *k*-means cluster, compared to all genes in our dataset, was calculated and plotted as log2 ratio in a heatmap.

## Calculation of gene expression noise

Gene expression noise at the mRNA and protein levels was calculated as the coefficient of variation (CV; standard deviation divided by the mean) of log2-transformed RPKM and SILAC ratios, respectively. To avoid dividing by zero (for unchanged genes with a log2 ratio of zero), a constant value of 10 was added to all mRNA and protein log2 ratios before calculating the noise.

## Calculating the clustering degree

To define local gene density in a manner that can be applied to both the sequence and the 3D structure of the genome, we determined the average distance of a gene to its three nearest neighbouring genes. We calculated three such "clustering degrees" for each gene in our dataset. For the sequence-based clustering degree, the distance to neighbouring genes was calculated in base pairs. For intrachromosomal clustering in 3D, gene distance was calculated based on Hi-C counts. However, we only considered "nearest" neighbours which were at least 500 kb away in terms of DNA sequence, to catch long-range interactions and avoid replicating the sequence-based clustering degree. For interchromosomal clustering, we considered the three nearest neighbours on other chromosomes, based on interchromosomal Hi-C contacts.

## Heterochromatin profiles of upstream regions

Chromatin states throughout the LCL genome were previously described (Ernst *et al*, 2011). To simplify the analysis, we combined the five inactive chromatin states defined by Ernst *et al* ("Heterochromatin", "Repressed", "Repetitive", "Poised Promoter" and "Insulator") into one "heterochromatin" state. We then scanned the promoter region of test genes for the presence of heterochromatin, moving in 100-bp intervals from $-50,000$ bp to $+10,000$ bp relative to their transcription start site.

## Calculating epigenetic similarity

Epigenetic similarity was calculated on the basis of the histone mark abundance within gene bodies (see section "Analysis of cluster features" for processing of ChIP-seq data). For this analysis, we considered H2AFZ, H3K4me1, H3K4me2, H3K4me3, H3K27ac, H3K79me2, H3K9ac, H3K9me3, H4K20me1, H3K27me3, H3K36me3 and CpG methylation, but not GC content, gene length and replication timing. For every pair of genes, we then determined how similar or dissimilar they are regarding the abundance of these epigenetic features. This was calculated using the Mahalanobis distance measure, which takes into account that some histone marks strongly covary.

## Analysis of post-transcriptional mechanisms

mRNA half-lives in seven different LCLs were previously reported (Duan *et al*, 2013). We first calculated the average half-life of each mRNA in these LCLs. We considered two mRNAs to have a similar stability if the half-life of the more stable one was < 1.5-fold longer than the less stable one. mRNA targets of human miRNAs were also described previously (Helwak *et al*, 2013). Ribosome occupancy profiles for the LCL cell line panel were recently published (Battle *et al*, 2015). We considered ribosome profiles for 57 LCLs and 4,033 genes for which we had matching mRNA and protein measurements. We calculated Pearson correlation coefficients (PCCs) for ribosome profiles between all gene pairs. Two genes were said to

have correlated ribosome profiles at PCC > 0.5 (BH-adjusted *P*-value < 0.001). Proteins subjected to non-exponential degradation in human RPE-1 cells were also described recently (McShane *et al*, 2016). Finally, protein sequence lengths were downloaded from Uniprot (UniProt Consortium, 2015).

### Human paralogous genes

Human gene duplicates were downloaded from ENSEMBL (Yates *et al*, 2016). We only considered paralogues with at least 25% sequence identity.

### General data processing and plotting

Data processing was performed in R (R Core Team, 2016), unless indicated otherwise. Plots were created using the ggplot2 package (Wickham, 2009).

**Expanded View** for this article is available online.

## Acknowledgements

This work was supported by the Wellcome Trust through a Senior Research Fellowship to JR (grant number 103139). The Wellcome Trust Centre for Cell Biology is supported by core funding from the Wellcome Trust (grant number 203149).

## Author contributions

PG analysed Hi-C contact frequencies between the genes in our dataset. GK and JR designed the study, analysed the data and wrote the paper.

## Conflict of interest

The authors declare that they have no conflict of interest.

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
