## [Review Process File · Molecular Systems Biology]

Pervasive co-expression of spatially proximal genes is buffered at the protein level

Georg Kustatscher, Piotr Grabowski & Juri Rappsilber

Corresponding authors: Georg Kustatscher & Juri Rappsilber, University of Edinburgh

Review timeline:

Submission date:	19 January 2017
Editorial Decision:	17 February 2017
Revision received:	05 June 2017
Editorial Decision:	26 June 2017
Revision received:	21 July 2017
Accepted:	24 July 2017

Editor: Maria Polychronidou

Transaction Report:

1st Editorial Decision

17 February 2017

Thank you again for submitting your work to Molecular Systems Biology. We have now heard back from the two referees who agreed to evaluate your manuscript. As you will see below, the reviewers appreciate that you address an interesting topic. They raise however a series of concerns, which should be carefully addressed in a major revision.

Without repeating all the points listed below, some of the more fundamental issues are the following:

- Additional analyses are required to examine whether the observed buffering at the protein level is due to a selective or a neutral process. Also, the extent at which the presented findings can be explained by the noise reduction model needs to be examined.
- Additional data (i.e. Ribo-Seq) and analyses are required in order to examine the mechanisms involved in post-transcriptional buffering of expression levels.

Of course, all other points raised by the referees would need to be convincingly addressed.

As we had previously discussed, we are flexible in terms of the format. Therefore, if you prefer to submit the manuscript as a (short) Article instead of a Report it is perfectly fine for us. In this case please keep in mind that the Results and Discussion need to be two separate sections.

On a more editorial level, we would like to ask you to address the following issues:

- Please provide a .doc file for the main manuscript text and individual files for the main figures.

- We recently replaced Supplementary Information by Expanded View (EV). In this format, a limited number (MAXIMUM 5) of Supplementary Figures are included in the article as EV figures. Expanded View Figures should be cited as 'Figure EV1', 'Figure EV2' etc. Please provide their legends in the main text and the illustrations as separate files. If you decide to not use the EV format, then all additional figures should be included in a 'traditional' supplementary PDF, now called the *Appendix*. Appendix figures should be labeled and cited as: "Appendix Figure S1" etc.

- Datasets should be labeled and cited as Dataset EV1, Dataset EV2 etc. Please provide the dataset description as a README.txt file, zipped together with the corresponding .csv file.

- Please provide a "standfirst text" summarizing the study in one or two sentences (approximately 250 characters), three to four "bullet points" highlighting the main findings and a "synopsis image" (211x157 pixels, jpeg format) to highlight the paper on our homepage.

- When you resubmit your manuscript, please download our CHECKLIST and include the completed form in your submission. *Please note* that the Author Checklist will be published alongside the paper as part of the transparent process

REFeree REPORTS

Reviewer #1:

Review of Kustatcher et al.

In this brief paper the authors return to the problem of genes in bidirectional orientation known to show strong co-expression at the mRNA level. Is this co-expression selectively favourable or deleterious? The team argue that if it is deleterious it should be buffered at the protein level. They find evidence for buffering. In a follow on analyses they show that there exist 3D genome compartments for mRNA expression but that protein similarity in expression is more to do with subcellular location than genome proximity. These 3D results support their view that protein co-expression is about functional interactions rather than genome proximity while mRNA co-expression is about chromatin level/3D incidental effects (but don't cite papers claiming otherwise[1]).

The team do reasonable job of introducing the literature but I think they miss a trick or two and omit some key literature - that predicts their results. Moreover, they never consider that the evidence for protein buffering might reflect a neutral process and one need not necessarily evoke the idea that this is selectively relevant. Thus, while I have few queries about what was done, I don't think their interpretation either gives credit where due nor does their evidence quite support their assertions. Both can be corrected via a modest rewrite. In a few places their results seem to contrast with prior (uncited) literature.

Prior literature:

In yeast similar analysis suggests that closely linked genes are co-expressed[2] but that this co-expression is functional for only approximately 2% of all such pairs. For these pairs alone do we see evidence for functional similarity[3] or conservation of gene order[4]. Rather most co-expression of closely linked genes can be attributed to chromatin dynamics [3]. This was I think the first claim that co-expression is not itself the product of selection. This 2% figure is striking close to the numbers in the present paper and this comparison should be made. Indeed, did anyone ever suggest that all co-expression was to enable functionally related genes to coordinate? Have we something of a straw man here?

However, the more important issue that the authors are missing is that bidirectional promoters have been argued to be device to reduce noise and that this model seems to predict their results (but is unmentioned). This noise reduction has as an incidental side effect some level of co-expression at the RNA level [5]. If you look at the toy model simulations in fig 2 of Wang's paper you see many

cases where independence is low (ie. genes are coupled) that causes low noise but high coexpression. This was argued to be especially important for housekeeping/essential genes as they are likely to be under selection for reduced noise as they are dosage sensitive (by definition essential genes are dosage sensitive). The authors found some protein level noise evidence for this. You might like also to consult Sandhu's work on this problem [6-8].

The current paper would seem to support this model rather well: only for the most highly co-expressed do we see functional coupling and for the others the mRNA level co-expression is a necessary statistical consequence of selection for reduced fluctuation (buffering). I think this context would need to be made more explicit, not least because recent data [9] indeed supports the more general view [10] that gene location is an important modulator of noise.

I find this model attractive as it predicts that many genes will be in close proximity often coupled to bidirectional promoters but that the utility of this is not to enable co-expression (except the 2%) - it is to enable reduced noise for noise sensitive genes. Wang finds much evidence at the protein level consistent with this (e.g. a gene in bidirectional orientation tends to be essential low noise and rarely subtelomeric - yeast's hotspot for noise generation). It makes for a parsimonious interpretation of CUTs driven off bidirectional promoters: their function is to reduce the noise of the (commonly essential) gene on the other strand.

Alternative model

Here I would like to play devil's advocate and suggest that dampening of protein co-expression need not imply presence of selection for dampening. First in Wang's simulations [5] it seems that protein co-expression is commonly much reduced from the level of mRNA co-expression simply owing to the extra processes - translation and protein decay. This suggests that an active control process is not needed (the had no selection for such protein level effects). What would be needed is an active process to enable protein co-expression for the special 2%. I think one can see intuitively why such passive buffering might go on. If you imagine that translatability of an mRNA is a stochastic variable drawn from a distribution with a wide variation then two mRNAs may easily be co-expressed, but the two mRNA may be processed at very different rates (which the RiboSeq data seems to support). Similarly, if protein decay rates vary this will destroy coexpression signals. Thus I don't think one can directly infer from the results that the co-expression at the mRNA level per se is deleterious. It could be a neutral by-product of selection for noise reduction, with the buffering achieved by no selective intervention.

I have a few other (related) comments:

Abstract: why presume that the co-expression is either advantageous or deleterious - why not neutral or a necessary byproduct of selection for reduced protein level noise?

Page 2 - you point out that some data suggests that co-expression can be deleterious. What then to make of Dai et al's analysis showing conservation of co-expression even after genome rearrangement [11]? This seems to directly contradict your assertion.

Page 2 - when discussing origination it would be worth mentioning that transgenes adopt the expression profile of the genomic neighbourhood they insert into (e.g. [12]) and that inserts can affect their neighbours - a problem in gene therapy [13].

Page 3 when looking at genes 50kb apart did you include the prior 167? The analyses should be independent. And how was "apart" defined - promoter to promoter, ATG to ATG, from nearest ends?

Page 4. What to make of the claim that long range 3D interactions are between functionally related genes [1]? Have you simply mis-defined functional similarity?

Page 7 discussion - you note that housekeeping genes cluster tending to prevent accidental silencing (this is the noise model) and that this makes them susceptible to interference. You could say that, or you could say the mRNA level interference and co-expression are necessary components of the avoidance of accidental silencing. As I see it (and I could well be wrong), active genes enable genes

in their neighbourhood to also be active so as to prevent any of them have too low a dose (which I think is sort of this model: [10]). Co-expression is an epiphenomenon of this. This fits well with Wang's noise model.

Page 12 I think you should reference Williams and Bowles somewhere [14]

Page 15 Last line of mRNA abundances section: "To do so, ..." I don't understand the normalization that is going on. Why subtract? Don't you want to normalize to ensure all libraries had the same mean or median? I am probably missing something.

1. Thevenin A, Ein-Dor L, Ozery-Flato M, Shamir R. Functional gene groups are concentrated within chromosomes, among chromosomes and in the nuclear space of the human genome. *Nucleic Acids Res.* 2014;42(15):9854-61. doi: 10.1093/nar/gku667. PubMed PMID: WOS:000343220300033.
2. Cohen BA, Mitra RD, Hughes JD, Church GM. A computational analysis of whole-genome expression data reveals chromosomal domains of gene expression. *Nature Genet.* 2000;26(2):183-6. PubMed PMID: ISI:000089638800014.
3. Batada NN, Urrutia AO, Hurst LD. Chromatin remodelling is a major source of coexpression of linked genes in yeast. *Trends Genet.* 2007;23(10):480-4. PubMed PMID: ISI:000250636600003.
4. Hurst LD, Williams EJ, Pal C. Natural selection promotes the conservation of linkage of co-expressed genes. *Trends Genet.* 2002;18(12):604-6. PubMed PMID: 12446137.
5. Wang GZ, Lercher MJ, Hurst LD. Transcriptional Coupling of Neighboring Genes and Gene Expression Noise: Evidence that Gene Orientation and Noncoding Transcripts Are Modulators of Noise. *Genome Biol Evol.* 2011;3:320-31. doi: 10.1093/gbe/evr025. PubMed PMID: ISI:000290252700030.
6. Sandhu KS. Did the modulation of expression noise shape the evolution of three dimensional genome organizations in eukaryotes? *Nucleus-Austin.* 2012;3(3):286-9. doi: 10.4161/nucl.20263. PubMed PMID: WOS:000315928100014.
7. Sandhu KS, Li GL, Poh HM, Quek YLK, Sia YY, Peh SQ, et al. Large-Scale Functional Organization of Long-Range Chromatin Interaction Networks. *Cell Reports.* 2012;2(5):1207-19. doi: 10.1016/j.celrep.2012.09.022. PubMed PMID: WOS:000314457700018.
8. Sandhu KS, Li GL, Sung WK, Ruan YJ. Chromatin Interaction Networks and Higher Order Architectures of Eukaryotic Genomes. *J Cell Biochem.* 2011;112(9):2218-21. doi: 10.1002/jcb.23155. PubMed PMID: WOS:000294769500004.
9. Chen X, Zhang J. The Genomic Landscape of Position Effects on Protein Expression Level and Noise in Yeast. *Cell Syst.* 2016;2(5):347-54. doi: 10.1016/j.cels.2016.03.009. PubMed PMID: 27185547; PubMed Central PMCID: PMC4882239.
10. Batada NN, Hurst LD. Evolution of chromosome organization driven by selection for reduced gene expression noise. *Nature Genet.* 2007;39(8):945-9. PubMed PMID: ISI:000248446900006.
11. Dai Z, Xiong Y, Dai X. Neighboring Genes Show Interchromosomal Colocalization after Their Separation. *Mol Biol Evol.* 2014;31(5):1166-72. doi: 10.1093/molbev/msu065.
12. Gierman HJ, Indemans MHG, Koster J, Goetze S, Seppen J, Geerts D, et al. Domain-wide regulation of gene expression in the human genome. *Genome Res.* 2007;17(9):1286-95. doi: 10.1101/gr.6276007.
13. Buckley RH. Gene therapy for SCID--a complication after remarkable progress. *The Lancet.* 2002;360(9341):1185-6.
14. Williams EJB, Bowles DJ. Coexpression of neighboring genes in the genome of *Arabidopsis thaliana*. *Genome Res.* 2004;14(6):1060-7. PubMed PMID: ISI:000221852400008.

Reviewer #2:

This manuscript addresses how domains of co-regulated genes maintain their optimal expression levels in face of transcriptional interference as an outcome of the regulation of neighboring genes. The authors propose an interesting notion of a buffering system to maintain the proper protein level of spatially close genes which are usually co-regulated at the transcription level. The authors investigate this by examining available data for mRNA and protein abundances of 60 human lymphoblastoid cell lines (LCLs). They concluded that, in general, co-regulation of close genes at mRNA level is not reflected by their protein abundances, especially in regard to non-functionally

related genes, suggesting that post-transcriptional events take place in order to buffer regulatory interference from nearby genes.

The work provides significant contribution to the current knowledge of gene co-regulation patterns in the human genome, especially due to innovative analyses matching mRNA and protein expression of the same samples (LCLs), and also by integrating publicly available genome compartment (Hi-C) and epigenomic information (histone marks). The major conclusion that co-regulation at protein and mRNA levels are not paralleled is supported by robust evidence. Despite this however, the manuscript does lack a proposed explanation of the mechanism of the buffering system: how the protein level be buffered given the co-regulated mRNA level. The authors should provide more results covering the major and minor comments, listed below, before making the decision whether the manuscript should be accept

Major comments:

- 1) In Fig S1 D&F, the results look like fold change between the genes in the study and other genes at the protein level is generally lower than at the mRNA level. Does it mean that the protein abundance is intrinsically and globally buffered to some extent? This might due to the different rates of RNA synthesis/degradation and protein synthesis/degradation. The authors need to explain how this buffering mechanism is different from the buffering mechanism discussed in this paper.
- 2) For the mRNA co-regulation across all the chromosomes, the corresponding chromatin interactions (Hi-C) should be added to help analyze if the co-regulation of mRNA is correlated with both intra and inter-chromosomal interaction. Related to Fig 1 D-F and Fig S4.
- 3) In Fig S3 and S4, the authors listed all the co-regulation events across chromosomes, but they do not discuss the most obvious co-regulation/buffering for example chr 16 and chr 19.
- 4) The authors should integrate more data, e.g. ribosome profiling, to further analyze the buffering mechanism discussed in the paper. In Fig 2 F, ribosome profiling was mentioned, but discussed limitedly. The authors should analyze if the mRNA co-regulated genes have similar ribosome binding rates, and carefully discuss how those differential ribosome binding profiles affect the buffering mechanism of protein co-regulation.
- 5) Although the authors recognize the limitation of considering mainly housekeeping genes due to the lack of comparable data, I question how general the conclusions are since ~21% (4188 / ~20000) of human protein coding genes were evaluated for co-regulation. How the datasets provided by The Human Protein Atlas (<http://www.proteinatlas.org/>) could be helpful to extend the analyses to more tissue/cell specific genes? Did the authors consider to analyze those datasets?
- 6) Previous work suggest that large variations (noise) in the expression of housekeeping genes are likely detrimental to cell fitness (e.g. Fraser, HB et al. Plos Biol., 2004, Newman, JRS et al. Nature 2006). Thus, I wonder how robust the buffering at protein level is to overcome the expression noise from regulatory interference of close genes. Is there a way to assess how expression noise relates to gene distance, and what is the extent of protein buffering regarding noisier genes (e.g. by using replicates)?
- 7) The authors emphasize that signals of co-regulation are linked to evolutionary origin of nearby genes rather than common biological functions. However, no evolutionary analyses were performed in regard to duplicated pairs, for example. How gene duplication (tandem and non-local) would bias the observations of co-regulation at mRNA and protein levels? Furthermore, I would suggest a more detailed discussion about the evolutionary perspective and consequences of clustered gene organization. There are valuable literature on this topic (see for instance Ghanbarian & Hurst, Mol. Biol. and Evol. 2015) that could help to greatly enrich the Discussion.

Minor points:

- 1) In page 3, "We next considered all gene pairs that are up to 50 kb apart, regardless of their orientation...". Please add the absolute numbers of these gene pairs of up to 50 kb. It would be even better to generate a supplementary table like Table S2.
- 2) For all the mRNA co-regulation and promoter interaction (Hi-C)-correlation analysis, the correlation factor should be analyzed and reflected on the figure. (Fig 1D&F, and Fig S4 after adding Hi-C data.)
- 3) In page 6, "For example, in comparison to T2, T1 gene bodies are enriched for H3K9me3, are longer, replicate later, and have a lower GC content. These differences mirror those observed between A2 and A1 subcompartments (Rao et al., 2014)." How about other epigenetic features, e.g.

H3K4me4, H3K27ac, etc, which represent the active histone markers? Do they also mirror the differences of A1 and A2 subcompartments?

4) Second results section (Genes with similar functions...): It was not clear how close the functionally related genes are. Was the analysis conducted for genes enclosed at specific distance? How far? An specific analysis could be: Do closer and functionally related genes have higher protein abundance correlations compared to those farther but also functionally related?

5) Last paragraph of third results section (Long-range co-regulation...): Please rewrite for clarification. The authors introduce the concept of genome compartments without explaining how they were defined. A rough explanation was provided just later in the last paragraph of the next section. I suggest to relocate it and improve the explanation.

6) First paragraph of fourth results section (Genes with co-regulated mRNAs...): It was not clear "As expected, genes from T1 and T2 are strongly enriched...". Please clarify "as expected".

7) Please clarify what "consensus expression" means and how it was calculated.

8) Avoid using words that give idea of intention. This is not meaningful in evolutionary terms. Some examples:

"...if that is the purpose or an undesired consequence..."

"...transcribed regions to acquire additional genes..."

"...similar unintended mRNA co-regulation..."

"...undesired consequence of genome evolution..."

Point-by-point response to the reviewers

“Pervasive regulatory interference between spatially close genes is buffered at the protein level” by Kustatscher, Grabowski and Rappsilber

We would like to thank the reviewers for their very constructive criticism, which helped us to expand and improve our manuscript considerably. We believe that we have now addressed their questions in full, and provide a point-by-point response below.

Reviewer #1:

In this brief paper the authors return to the problem of genes in bidirectional orientation known to show strong co-expression at the mRNA level. Is this co-expression selectively favourable or deleterious? The team argue that if it is deleterious it should be buffered at the protein level. They find evidence for buffering. In a follow on analyses they show that there exist 3D genome compartments for mRNA expression but that protein similarity in expression is more to do with subcellular location than genome proximity. These 3D results support their view that protein co-expression is about functional interactions rather than genome proximity while mRNA co-expression is about chromatin level/3D incidental effects (but don't cite papers claiming otherwise[1]).

We would like to thank the reviewer for pointing out several important studies that we failed to cite. We have to admit that, being new to the field, many of these were not known to us before, including reference [1] (Thévenin *et al*, 2014). In a nutshell, Thévenin *et al*. demonstrate that genes with similar functions are often found in close proximity in the human genome, both in terms of intra-chromosomal and 3D nuclear distance. At first sight, this would appear to contradict our findings that the majority of closeby, coexpressed transcripts are not functionally related and therefore not coexpressed on the protein level.

However, the two results are not mutually exclusive, and we actually share Thévenin's observation, but only alluded to it very briefly in the description of Fig 1B. This figure shows that 3% of closeby genes (TSSs within 50 kb) have co-regulated protein abundances, i.e. probably share a common function. This is significantly more than expected by random chance. In other words, like Thévenin *et al*. we see a tendency of functionally related genes to be located close to one another.

The difference is that we also observe that coexpression of mRNAs affects 22% of closeby gene pairs, far more than the ~3% that are functionally related. Therefore, while genes with common functions are indeed more likely to co-localise

in the genome, co-function does not underlie the mRNA coexpression of the *majority* of closeby genes. Please note that Thévenin *et al.* analyse the relationship between gene function and gene position but do not analyse gene expression levels.

To clarify this point, we have inserted a short new Results section (“A fraction of closeby genes is enriched for similar functions”; page 5) and the new Figure EV2. We reproduced the findings from Thévenin *et al.* using our dataset and confirmed that spatially close genes, either those closeby in sequence or those sharing the same HiC subcompartment in 3D, are more likely to have a common function than gene pairs which are further apart. However, we also show that co-function can only explain a fraction of the observed mRNA co-regulations.

The team do reasonable job of introducing the literature but I think they miss a trick or two and omit some key literature - that predicts their results. Moreover, they never consider that the evidence for protein buffering might reflect a neutral process and one need not necessarily evoke the idea that this is selectively relevant. Thus, while I have few queries about what was done, I don't think their interpretation either gives credit where due nor does their evidence quite support their assertions. Both can be corrected via a modest rewrite. In a few places their results seem to contrast with prior (uncited) literature.

Please see below for responses regarding buffering mechanisms and additional literature.

Prior literature:

In yeast similar analysis suggests that closely linked genes are co-expressed [2] but that this co-expression is functional for only approximately 2% of all such pairs. For these pairs alone do we see evidence for functional similarity [3] or conservation of gene order [4]. Rather most co-expression of closely linked genes can be attributed to chromatin dynamics [3]. This was I think the first claim that co-expression is not itself the product of selection. This 2% figure is striking close to the numbers in the present paper and this comparison should be made. Indeed, did anyone ever suggest that all co-expression was to enable functionally related genes to coordinate? Have we something of a straw man here?

We believe that, in general, coexpression is still seen as a “strong indicator of functional associations” (Szklarczyk *et al*, 2015). In fact, we would argue that this interpretation is correct, except that it should be restricted to protein coexpression.

We now cite these references and discuss the striking agreement in the number of the identified coexpression incidents that are functionally relevant. This is particularly interesting given that references [2-4] describe work in yeast, whose genome organisation can be quite different from mammals. For example,

coexpression of adjacent genes is more commonly seen for tandem rather than bidirectional gene pairs in yeast (Kruglyak & Tang, 2000; Batada *et al*, 2007) and yeast chromosomes are smaller than the HiC subcompartments we analyse.

We further built on this pioneering work and investigated the potential mechanisms causing coexpression of neighbouring genes. This is described in the new Results section “Coexpression of closeby genes is driven by stochastic epigenetic fluctuations and regulatory interference” (page 10) and the new Figure 3. In short, we considered both a passive chromatin fluctuation model (Raj *et al*, 2006; Batada *et al*, 2007) as well as direct transcriptional interference (Wang *et al*, 2011), and find evidence that both exist in humans.

However, the more important issue that the authors are missing is that bidirectional promoters have been argued to be device to reduce noise and that this model seems to predict their results (but is unmentioned). This noise reduction has as an incidental side effect some level of co-expression at the RNA level [5]. If you look at the toy model simulations in fig 2 of Wang's paper you see many cases where independence is low (ie. genes are coupled) that causes low noise but high coexpression. This was argued to be especially important for housekeeping/essential genes as they are likely to be under selection for reduced noise as they are dosage sensitive (by definition essential genes are dosage sensitive). The authors found some protein level noise evidence for this. You might like also to consult Sandhu's work on this problem [6-8].

We agree that the noise-reduction effect of bidirectional promoters (Wang *et al*, 2011) [5] and gene clustering in general (Batada & Hurst, 2007) could explain why functionally unrelated housekeeping genes tend to cluster in the first place. The idea that mRNA coexpression of neighbouring genes is an incidental side effect of noise-reduction fits perfectly with our observation that such non-functional coexpression is buffered at the protein level. We have re-written multiple sections of the manuscript to better incorporate this idea. In addition, we tested if the observations from Batada and Hurst are supported in the context of human LCLs, which we find to be the case (new Fig 3A).

Sandhu's work focusses on the relationship between 3D genome structure and noise reduction, and finds that genes with more interchromosomal contacts actually have higher expression noise. To resolve this apparent discrepancy, we also analyse mRNA and protein expression noise in a 3D context. We find that long-range intrachromosomal contacts (> 500 kb) reduce expression noise, but confirm Sandhu's data that interchromosomal contacts have the opposite effect (new Fig 3A).

The current paper would seem to support this model rather well: only for the most highly co-expressed do we see functional coupling and for the others the mRNA level co-expression is a necessary statistical consequence of selection for reduced fluctuation (buffering). I think this context would need to be made

more explicit, not least because recent data [9] indeed supports the more general view [10] that gene location is an important modulator of noise.

I find this model attractive as it predicts that many genes will be in close proximity often coupled to bidirectional promoters but that the utility of this is not to enable co-expression (except the 2%) - it is to enable reduced noise for noise sensitive genes. Wang finds much evidence at the protein level consistent with this (e.g. a gene in bidirectional orientation tends to be essential low noise and rarely subtelomeric - yeast's hotspot for noise generation). It makes for a parsimonious interpretation of CUTs driven off bidirectional promoters: their function is to reduce the noise of the (commonly essential) gene on the other strand.

We agree with these interpretations and have re-written multiple sections of the manuscript to emphasise the connection between noise-reduction and coexpression. We also now discuss the references (Chen & Zhang, 2016) [9] and (Batada & Hurst, 2007) [10].

Alternative model

Here I would like to play devil's advocate and suggest that dampening of protein co-expression need not imply presence of selection for dampening. First in Wang's simulations [5] it seems that protein co-expression is commonly much reduced from the level of mRNA co-expression simply owing to the extra processes - translation and protein decay. This suggests that an active control process is not needed (they had no selection for such protein level effects). What would be needed is an active process to enable protein co-expression for the special 2%. I think one can see intuitively why such passive buffering might go on. If you imagine that translatability of an mRNA is a stochastic variable drawn from a distribution with a wide variation then two mRNAs may easily be co-expressed, but the two mRNA may be processed at very different rates (which the RiboSeq data seems to support). Similarly, if protein decay rates vary this will destroy coexpression signals. Thus I don't think one can directly infer from the results that the co-expression at the mRNA level per se is deleterious. It could be a neutral by-product of selection for noise reduction, with the buffering achieved by no selective intervention.

Indeed, non-selective buffering may be an alternative explanation to an active post-transcriptional mechanism that specifically targets coexpressed neighbouring genes. Now we systematically analysed the involvement of various steps of post-transcriptional regulation, such as mRNA stability, translation and protein degradation. As suggested by the reviewer, we find no evidence that any of these steps selectively target coexpressed neighbouring genes. In contrast, gene pairs showing coexpression also at the protein level (the 2%) are much more likely to be subjected to similar post-transcriptional regulation, such as having correlated ribosome profiles. These data are summarised in the new Figure 4. We conclude

and discuss that buffering of co-regulated neighbouring genes is a passive process, resulting simply from a lack of coordination between post-transcriptional mechanisms.

I have a few other (related) comments:

Abstract: why presume that the co-expression is either advantageous or deleterious - why not neutral or a necessary byproduct of selection for reduced protein level noise?

Page 2 - you point out that some data suggests that co-expression can be deleterious. What then to make of Dai *et al*'s analysis showing conservation of co-expression even after genome rearrangement [11]? This seems to directly contradict your assertion.

We agree that co-expression of mRNAs does not have to be deleterious and removed these statements from the paper. Overall, there appears to be somewhat conflicting evidence regarding the conservation of expression clusters. Now we expanded the discussion of the evolutionary aspects of co-expression of gene neighbours, and moved it into the "Discussion" section of the manuscript.

With regards to Dai *et al* [11], we think this could be a similar "point-of-view" problem as with the functional similarity of clustered genes, i.e. significant enrichment does not necessarily provide an explanation for the bulk of the observations. Dai *et al* show that gene pairs that were separated during evolution are more likely to co-localise in 3D than random gene pairs. However, they find such co-localisation in 3D only for 5.6% of their tested pairs. These gene pairs are also more likely to be bound by the same transcription factors, so it is possible that they belong to a small subset of genes for which co-localisation is actually important for their regulation.

Notably, the conservation of coexpression is only a small aspect of Dai *et al*'s paper. It is unclear how many of the 5.6% genes whose spatial proximity is conserved also have conserved mRNA coexpression. They show that separated genes co-localising in 3D are more strongly coexpressed than separated genes that do *not* co-localise, which is in line with our report. However, this does not really address the "conservation of coexpression" question. For that, we would need to know how many of the separated gene pairs were once coexpressed and are still coexpressed. Taken together, we do not think that it can be concluded from Dai *et al*'s analysis that coexpressed, linked gene pairs generally maintain their proximity (in 3D) and their coexpression after separation (they also do not make that claim).

We have cited Dai *et al* now to point out to the reader that separated neighbouring genes have a tendency to show inter-chromosomal localisation. However, we think it would be confusing to discuss the issue of coexpression maintenance in this context at this point.

Page 2 - when discussing origination it would be worth mentioning that transgenes adopt the expression profile of the genomic neighbourhood they insert into (e.g. [12]) and that inserts can affect their neighbours - a problem in gene therapy [13].

We now mention that transgenes adopt expression and noise profiles of their insertion site, citing the publications from Chen and Zhang [9] and Gierman *et al* [12]. While no doubt this can be a problem in gene therapy, as far as we can tell the particular example described by Buckley [13] appears to refer to a different issue, namely the disruption of a tumor suppressor gene by transgene insertion.

Page 3 when looking at genes 50kb apart did you include the prior 167? The analyses should be independent. And how was "apart" defined - promoter to promoter, ATG to ATG, from nearest ends?

Yes, we originally included the 167 bidirectional pairs among the gene pairs that are 50 kb apart. We changed this now such that the analyses are independent. Notably, the change is so subtle that it is not visible in the updated Fig 1B and the (rounded) percentages in the main text remain the same. The difference between closeby and random gene pairs is now slightly less significant, as the P value had to be updated from 6×10^{-15} to 4×10^{-14} . We also clarified that "apart" refers to the distance between the transcription start sites.

Page 4. What to make of the claim that long range 3D interactions are between functionally related genes [1]? Have you simply mis-defined functional similarity?

Please see our answer to the first question. In short, as we show with our new figure EV2, we also find significant enrichment of functional similarity among genes that show long range 3D interactions. However, this explains only a very small fraction of interacting gene pairs, so long range 3D interactions are not between functionally related genes *in general*.

We used four different ways to describe functional similarity in this paper. Where accuracy was important, we relied on subunits of the same protein complex, enzymes of the same metabolic pathway, or proteins with identical subcellular localisation. Where it was important to be comprehensive, we relied on the STRING database, which compiles functional associations based on many different evidence types.

Page 7 discussion - you note that housekeeping genes cluster tending to prevent accidental silencing (this is the noise model) and that this makes them susceptible to interference. You could say that, or you could say the mRNA level interference and co-expression are necessary components of the avoidance of accidental silencing. As I see it (and I could well be wrong), active genes enable

genes in their neighbourhood to also be active so as to prevent any of them have too low a dose (which I think is sort of this model: [10]). Co-expression is an epiphenomenon of this. This fits well with Wang's noise model.

We agree with this interpretation and have changed our discussion accordingly.

Page 12 I think you should reference Williams and Bowles somewhere [14]

We now reference it in the introduction and on page 6 when discussing the lack of functional explanation for coexpressed neighbouring genes.

Page 15 Last line of mRNA abundances section: "To do so, ..." I don't understand the normalization that is going on. Why subtract? Don't you want to normalize to ensure all libraries had the same mean or median? I am probably missing something.

We subtract the log2 RPKMs of the reference cell line from those of the other LCLs. This is the equivalent of dividing the RPKMs by those of the reference line in non-log space, thereby calculating fold-changes. Median-normalisation was carried out as part of the correlation analysis (described in the later Method's section "Testing gene pairs for co-regulation").

1. Thevenin A, Ein-Dor L, Ozery-Flato M, Shamir R. Functional gene groups are concentrated within chromosomes, among chromosomes and in the nuclear space of the human genome. *Nucleic Acids Res.* 2014;42(15):9854-61. doi: 10.1093/nar/gku667. PubMed PMID: WOS:000343220300033.
2. Cohen BA, Mitra RD, Hughes JD, Church GM. A computational analysis of whole-genome expression data reveals chromosomal domains of gene expression. *Nature Genet.* 2000;26(2):183-6. PubMed PMID: ISI:000089638800014.
3. Batada NN, Urrutia AO, Hurst LD. Chromatin remodelling is a major source of coexpression of linked genes in yeast. *Trends Genet.* 2007;23(10):480-4. PubMed PMID: ISI:000250636600003.
4. Hurst LD, Williams EJ, Pal C. Natural selection promotes the conservation of linkage of co-expressed genes. *Trends Genet.* 2002;18(12):604-6. PubMed PMID: 12446137.
5. Wang GZ, Lercher MJ, Hurst LD. Transcriptional Coupling of Neighboring Genes and Gene Expression Noise: Evidence that Gene Orientation and Noncoding Transcripts Are Modulators of Noise. *Genome Biol Evol.* 2011;3:320-31. doi: 10.1093/gbe/evr025. PubMed PMID: ISI:000290252700030.
6. Sandhu KS. Did the modulation of expression noise shape the evolution of three dimensional genome organizations in eukaryotes? *Nucleus-Austin.* 2012;3(3):286-9. doi: 10.4161/nucl.20263. PubMed PMID: WOS:000315928100014.
7. Sandhu KS, Li GL, Poh HM, Quek YLK, Sia YY, Peh SQ, et al. Large-Scale Functional Organization of Long-Range Chromatin Interaction Networks. *Cell*

- Reports. 2012;2(5):1207-19. doi: 10.1016/j.celrep.2012.09.022. PubMed PMID: WOS:000314457700018.
8. Sandhu KS, Li GL, Sung WK, Ruan YJ. Chromatin Interaction Networks and Higher Order Architectures of Eukaryotic Genomes. *J Cell Biochem.* 2011;112(9):2218-21. doi: 10.1002/jcb.23155. PubMed PMID: WOS:000294769500004.
9. Chen X, Zhang J. The Genomic Landscape of Position Effects on Protein Expression Level and Noise in Yeast. *Cell Syst.* 2016;2(5):347-54. doi: 10.1016/j.cels.2016.03.009. PubMed PMID: 27185547; PubMed Central PMCID: PMC4882239.
10. Batada NN, Hurst LD. Evolution of chromosome organization driven by selection for reduced gene expression noise. *Nature Genet.* 2007;39(8):945-9. PubMed PMID: ISI:000248446900006.
11. Dai Z, Xiong Y, Dai X. Neighboring Genes Show Interchromosomal Colocalization after Their Separation. *Mol Biol Evol.* 2014;31(5):1166-72. doi: 10.1093/molbev/msu065.
12. Gierman HJ, Indemans MHG, Koster J, Goetze S, Seppen J, Geerts D, et al. Domain-wide regulation of gene expression in the human genome. *Genome Res.* 2007;17(9):1286-95. doi: 10.1101/gr.6276007.
13. Buckley RH. Gene therapy for SCID--a complication after remarkable progress. *The Lancet.* 2002;360(9341):1185-6.
14. Williams EJB, Bowles DJ. Coexpression of neighboring genes in the genome of *Arabidopsis thaliana*. *Genome Res.* 2004;14(6):1060-7. PubMed PMID: ISI:000221852400008.

Reviewer #2:

This manuscript addresses how domains of co-regulated genes maintain their optimal expression levels in face of transcriptional interference as an outcome of the regulation of neighboring genes. The authors propose an interesting notion of a buffering system to maintain the proper protein level of spatially close genes which are usually co-regulated at the transcription level. The authors investigate this by examining available data for mRNA and protein abundances of 60 human lymphoblastoid cell lines (LCLs). They concluded that, in general, co-regulation of close genes at mRNA level is not reflected by their protein abundances, especially in regard to non-functionally related genes, suggesting that post-transcriptional events take place in order to buffer regulatory interference from nearby genes.

The work provides significant contribution to the current knowledge of gene co-regulation patterns in the human genome, especially due to innovative analyses matching mRNA and protein expression of the same samples (LCLs), and also by integrating publicly available genome compartment (Hi-C) and epigenomic information (histone marks). The major conclusion that co-regulation at protein

and mRNA levels are not paralleled is supported by robust evidence. Despite this however, the manuscript does lack a proposed explanation of the mechanism of the buffering system: how the protein level be buffered given the co-regulated mRNA level. The authors should provide more results covering the major and minor comments, listed below, before making the decision whether the manuscript should be accept

We have now investigated the nature of the buffering mechanism, by analysing five different aspects of post-transcriptional gene regulation: mRNA stability, miRNA targeting, ribosome occupancy, protein degradation and the impact of coding length. These results point to buffering being a passive / neutral process. The subset of genes whose co-regulation is sustained at the protein level have a selectively higher coordination of post-transcriptional control. For example, their ribosome profiles tend to be better correlated. These data are summarised in the new Figure 4 and indicate that mRNA coexpression of closeby genes is buffered simply because their post-transcriptional regulation is not coordinated.

Major comments:

1) In Fig S1 D&F, the results look like fold change between the genes in the study and other genes at the protein level is generally lower than at the mRNA level. Does it mean that the protein abundance is intrinsically and globally buffered to some extent? This might due to the different rates of RNA synthesis/degradation and protein synthesis/degradation. The authors need to explain how this buffering mechanism is different from the buffering mechanism discussed in this paper.

It is true that fluctuations in mRNA abundances are generally dampened at the protein level. In the introduction we refer to a review on this topic (Liu *et al*, 2016), and also cite relevant primary publications. This aspect can also be seen in the new Figure 3A, where the gene expression noise (i.e. the expression variation) is weaker on the protein than the mRNA level. We show here that spatial proximity between genes is a major source of non-functional gene expression regulation, but that this is not propagated to the protein level. Although we originally anticipated to find a mechanism that specifically targets coexpressed, closeby genes and “disentangles” their coexpression, our new data show that this is indeed a passive process. In so far, it is not different to buffering of expression levels in response to, for example, a gene copy number change. Rather, as our new data suggest, the active and selective process is the coordination of post-transcriptional mechanisms that enables functionally related genes to still be coexpressed at the protein level.

2) For the mRNA co-regulation across all the chromosomes, the corresponding chromatin interactions (Hi-C) should be added to help analyze if the co-

regulation of mRNA is correlated with both intra and inter-chromosomal interaction. Related to Fig 1 D-F and Fig S4.

We have replaced Fig S4 with the new figures EV3 and EV4. The original figure S4 compared mRNA and protein correlations between all ~4,000 genes in our dataset. We could not simply add the HiC data to this plot, since intra- and interchromosomal HiC contacts are normalised differently and therefore not directly comparable. Also, we felt that comparing all ~4,000 genes at once made it difficult to visually identify co-regulation patches in the figure. To overcome these problems, we now split the figure into intra-chromosomal contacts (Fig EV3) and inter-chromosomal contacts (EV4), and for each case selected 3 representative examples that we present at high resolution.

3) In Fig S3 and S4, the authors listed all the co-regulation events across chromosomes, but they do not discuss the most obvious co-regulation/buffering for example chr 16 and chr 19.

These two chromosomes are now specifically shown in Fig EV3 and we mention that peculiar case of chromosome 19 in the main text (page 7). In short, chr 16 is one of the chromosomes with strongly defined mRNA co-regulation patches, similar to chr 11 that we show in Fig 1. Chromosome 19, on the other hand, is a very unusual case. It is one of the smallest human chromosomes, but also the one with the highest gene density. Essentially the entire chromosome appears to be a single mRNA co-regulation patch.

4) The authors should integrate more data, e.g. ribosome profiling, to further analyze the buffering mechanism discussed in the paper. In Fig 2 F, ribosome profiling was mentioned, but discussed limitedly. The authors should analyze if the mRNA co-regulated genes have similar ribosome binding rates, and carefully discuss how those differential ribosome binding profiles affect the buffering mechanism of protein co-regulation.

As mentioned above, we re-analysed the ribosome profiling data, together with four other aspects of post-transcriptional expression control (new Figure 4). We also added the new Result section “Buffering of non-functional mRNA co-expression is a non-selective process”. For example, about 10% of all co-regulated mRNA pairs also have correlated ribosome profiles. This is also true for those pairs where mRNA co-regulation is buffered at the protein level. In contrast, we see correlated ribosome profiles for about 30% of the pairs for which co-regulation is sustained at the protein level. This suggests selective post-transcriptional co-regulation of functionally related genes, rather than selective buffering of functionally unrelated ones.

5) Although the authors recognize the limitation of considering mainly housekeeping genes due to the lack of comparable data, I question how general the conclusions are since ~21% (4188 / ~20000) of human protein coding genes were evaluated for co-regulation. How the datasets provided by The Human Protein Atlas (<http://www.proteinatlas.org/>) could be helpful to extend the analyses to more tissue/cell specific genes? Did the authors consider to analyze those datasets?

Given that about half of all human genes are considered to be “housekeeping” according to the Human Protein Atlas (HPA), we cover about ~40% of them here (4,188 / ~10,000). So our analysis should be representative for these type of genes.

We did consider the HPA data. They cover a large number of tissues and cell lines with RNA-seq data, which we used in Figure S1. Unfortunately, the HPA does not contain matching quantitative proteomics data, but immunofluorescence data that we cannot use for this type of analysis. What we would need to extend this analysis to tissue-specific genes is a dataset where we have mRNA and protein abundances in different tissues across many individuals, and we are not aware of any such project.

However, small-scale studies have shown that transcriptional activation of induced genes (Spitz *et al*, 2003; Ebisuya *et al*, 2008) can lead to co-induction / co-activation of their genomic neighbours, even if these are not functionally related. It remains to be seen if this co-activation is restricted to the mRNA level.

6) Previous work suggest that large variations (noise) in the expression of housekeeping genes are likely detrimental to cell fitness (e.g. Fraser, HB *et al*. Plos Biol., 2004, Newman, JRS *et al*. Nature 2006). Thus, I wonder how robust the buffering at protein level is to overcome the expression noise from regulatory interference of close genes. Is there a way to assess how expression noise relates to gene distance, and what is the extent of protein buffering regarding noisier genes (e.g. by using replicates)?

The Hurst group has shown that gene clustering is associated with noise reduction, presumably because actively transcribed, neighbouring genes mutually reinforce their active state and reduce the likelihood that a gene is silenced by stochastic chromatin fluctuations. Most of the work in this area has so far been done in yeast, so we have now included an analysis of the expression noise in the human LCLs. As previously reported, we see that gene clustering reduces expression noise (new Fig 3A), so there is a relationship between noise and gene distance.

Expression noise, defined as the coefficient of variation between individual cells or, in our case, individual LCLs, exists both at the mRNA and the protein level. However, mRNAs are noisier than protein abundances (see Fig 3A for our data in human or Chen and Zhang, 2016, for data in yeast). One might argue that the expression noise itself is not completely buffered at the protein level. What is

completely buffered is the correlation of the mRNA fluctuations between neighbouring genes, i.e. the coexpression.

7) The authors emphasize that signals of co-regulation are linked to evolutionary origin of nearby genes rather than common biological functions. However, no evolutionary analyses were performed in regard to duplicated pairs, for example. How gene duplication (tandem and non-local) would bias the observations of co-regulation at mRNA and protein levels? Furthermore, I would suggest a more detailed discussion about the evolutionary perspective and consequences of clustered gene organization. There are valuable literature on this topic (see for instance Ghanbarian & Hurst, *Mol. Biol. and Evol.* 2015) that could help to greatly enrich the Discussion.

We expanded our discussion of gene order evolution considerably and moved it from the Introduction into the Discussion section. We also performed an analysis of duplicated gene pairs, based on paralogues defined by Ensembl. In principle, duplicated gene pairs could explain why some neighbouring genes are functionally related and coexpressed on both the mRNA and protein level. Indeed, we find that neighbouring genes with sustained protein coexpression are strongly enriched for duplicated gene pairs. However, as we discuss on page 14, gene duplication can only explain approximately a third of the functionally relevant coexpression of genomic neighbours.

Minor points:
1) In page 3, "We next considered all gene pairs that are up to 50 kb apart, regardless of their orientation...". Please add the absolute numbers of these gene pairs of up to 50 kb. It would be even better to generate a supplementary table like Table S2.

We added the absolute number of these gene pairs (929) to the main text. In addition, we now provide Table EV2, which contains the mRNA and protein correlation for these gene pairs and the bidirectional ones.

2) For all the mRNA co-regulation and promoter interaction (Hi-C)-correlation analysis, the correlation factor should be analyzed and reflected on the figure. (Fig 1D&F, and Fig S4 after adding Hi-C data.)

For every HiC map in the manuscript (Fig 1F, Fig EV3, EV4) we now also report how well it correlates with the respective mRNA and protein co-regulation maps. In addition, the new Appendix Table S2 reports the mRNA - HiC correlation factor for every chromosome.

3) In page 6, "For example, in comparison to T2, T1 gene bodies are enriched for H3K9me3, are longer, replicate later, and have a lower GC content. These differences mirror those observed between A2 and A1 subcompartments (Rao et al., 2014) ." How about other epigenetic features, e.g. H3K4me4, H3K27ac, etc, which represent the active histone markers? Do they also mirror the differences of A1 and A2 subcompartments?

Yes they do, so we updated the sentence to reflect that.

4) Second results section (Genes with similar functions...): It was not clear how close the functionally related genes are. Was the analysis conducted for genes enclosed at specific distance? How far? An specific analysis could be: Do closer and functionally related genes have higher protein abundance correlations compared to those farther but also functionally related?

This paragraph and the accompanying figure (now called Fig EV1) was intended as a quality control for the proteomics data. Until that point we observe frequent co-regulation of mRNAs, but not of proteins. One might argue that quantitation by RNA sequencing could potentially be more accurate than proteomics, so rather than observing a buffering effect we might simply fail to detect protein co-regulation. Therefore, the analysis in this paragraph is not restricted to genes at any specific distance, but aims to assess if protein co-regulation can be detected at all. We now emphasize that no distance constraint was used in this section.

5) Last paragraph of third results section (Long-range co-regulation...): Please rewrite for clarification. The authors introduce the concept of genome compartments without explaining how they were defined. A rough explanation was provided just later in the last paragraph of the next section. I suggest to relocate it and improve the explanation.

We now explain how genome compartments and subcompartments were defined, immediately after we first mention them in the text.

6) First paragraph of fourth results section (Genes with co-regulated mRNAs...): It was not clear "As expected, genes from T1 and T2 are strongly enriched...". Please clarify "as expected".

Indeed, this was not really expected, so we removed the "as expected".

7) Please clarify what "consensus expression" means and how it was calculated.

“Consensus expression” referred to the median fold-change of the genes in each k -means cluster, calculated for each of the 59 LCLs relative to the reference LCL. However, in an effort to make the manuscript easier to read and avoid unnecessary figure panels, we removed this section now. Its purpose was to show that the expression changes of genes belonging to the “T1” k -means cluster are anti-correlated with those belonging to the “T2” cluster, as well as the fact that mRNA co-regulation clusters are not reflected at the protein level and vice versa. That same information was conveyed by Supplementary Figure S6 in a simpler way, so we used that figure as Fig 2B instead.

8) Avoid using words that give idea of intention. This is not meaningful in evolutionary terms. Some examples:

“...if that is the purpose or an undesired consequence...”

“...transcribed regions to acquire additional genes...”

“...similar unintended mRNA co-regulation...”

“...undesired consequence of genome evolution...”

These phrases were removed and we tried to avoid making further statements of intention.

REFERENCES

- Batada NN & Hurst LD (2007) Evolution of chromosome organization driven by selection for reduced gene expression noise. *Nat. Genet.* **39**: 945–949
- Batada NN, Urrutia AO & Hurst LD (2007) Chromatin remodelling is a major source of coexpression of linked genes in yeast. *Trends Genet.* **23**: 480–484
- Chen X & Zhang J (2016) The Genomic Landscape of Position Effects on Protein Expression Level and Noise in Yeast. *Cell Syst* **2**: 347–354
- Ebisuya M, Yamamoto T, Nakajima M & Nishida E (2008) Ripples from neighbouring transcription. *Nat. Cell Biol.* **10**: 1106–1113
- Kruglyak S & Tang H (2000) Regulation of adjacent yeast genes. *Trends Genet.* **16**: 109–111
- Liu Y, Beyer A & Aebersold R (2016) On the Dependency of Cellular Protein Levels on mRNA Abundance. *Cell* **165**: 535–550
- Raj A, Peskin CS, Tranchina D, Vargas DY & Tyagi S (2006) Stochastic mRNA synthesis in mammalian cells. *PLoS Biol.* **4**: e309
- Spitz F, Gonzalez F & Duboule D (2003) A global control region defines a chromosomal regulatory landscape containing the HoxD cluster. *Cell* **113**: 405–417

- Szklarczyk D, Franceschini A, Wyder S, Forslund K, Heller D, Huerta-Cepas J, Simonovic M, Roth A, Santos A, Tsafou KP, Kuhn M, Bork P, Jensen LJ & von Mering C (2015) STRING v10: protein-protein interaction networks, integrated over the tree of life. *Nucleic Acids Res.* **43**: D447–52
- Thévenin A, Ein-Dor L, Ozery-Flato M & Shamir R (2014) Functional gene groups are concentrated within chromosomes, among chromosomes and in the nuclear space of the human genome. *Nucleic Acids Res.* **42**: 9854–9861
- Wang G-Z, Lercher MJ & Hurst LD (2011) Transcriptional coupling of neighboring genes and gene expression noise: evidence that gene orientation and noncoding transcripts are modulators of noise. *Genome Biol. Evol.* **3**: 320–331

Thank you again for submitting your work to Molecular Systems Biology. We have now heard back from the two referees who were asked to evaluate your study. As you will see below, both reviewers are satisfied with the modifications made. They raise however a couple of minor issues, which we would ask you to address in a minor revision.

We would also like to ask you to fix a few remaining editorial issues listed below:

- We would propose the modified title: "Pervasive co-expression of spatially proximal genes is buffered at the protein level".
- Since you have four figures we would suggest publishing the study as an Article (Reports typically have maximum three figures).
- Please include a Table of Contents in the beginning of the Appendix.
- Since the EV Tables are rather long, we would ask you to label them and cite them in the text as EV Datasets (Dataset EV1, Dataset EV2, Dataset EV3). Please provide the description of each dataset as a README.txt file together with the corresponding Dataset .csv file in a .zip folder.
- I have slightly edited the synopsis text (attached below). Could you please let me know if the text is OK or if you would like to make any further changes?
- I would also like to ask you to provide a larger version of the synopsis image (550 px (width) x 250-400 px (height)).

REFEREE REPORTS

Reviewer #1:

I liked this revised version very much. The major problems that I saw before were of interpretation (e.g. there is no evidence for selection buffering). The revision places the work better in to context. I also very much liked the new analysis of the stages of buffering which is helpful. In terms of references relating insertion site to gene expression they team should mention:

Akhtar W, de Jong J, Pindyurin AV, Pagie L, Meuleman W, de Ridder J, Berns A, Wessels LF, van Lohuizen M., van Steensel B. Chromatin position effects assayed by thousands of reporters integrated in parallel (2013). *Cell* 154(4):914-27

This finds evidence that transgenes adopt the local expression profile and can sometimes influence the expression of neighbours. A very nice experiment.

Reviewer #2:

The revised manuscript has detailed response in the revised version. The authors added a lot more results to better support the main conclusion of the paper. In general the revised manuscript should be accepted for publication but with one point should be revised:

On page 12, "Buffering of non-functional mRNA co-expression is a non-selective process" The title of this section seems over-claimed. I appreciate that the authors used several posttranscriptional control processes to support that functional-related co-expressed mRNA sustained at the protein level, but claiming that "non-functional mRNA co-expression is a non-selective process" seems to have no support in terms of direct evidence. In particular, the authors do not discuss any potential "selective process that specifically targets closeby genes and disentangles their expression patterns".

It seems more appropriate to claim rather the "buffering of non-functional mRNA co-expression tends to be a non-selective process". Other related claims in this section and the Discussion should be revised as well.

2nd Revision - authors' response

21 July 2017

RAPPSILBER
LABORATORY

WELLCOME TRUST CENTRE for CELL BIOLOGY

Juri Rappsilber, Ph.D.

Professor of Proteomics and

Senior Research Fellow of the Wellcome Trust

Institute of Cell Biology

The University of Edinburgh

Michael Swann Building

Edinburgh EH9 3BF

Scotland UK

Tel +44 (0) 131 651 7056

juri.rappsilber@ed.ac.uk

21 July 2017

Re: Pervasive co-expression of spatially proximal genes is buffered at the protein level

Dear Maria,

Thank you very much for your positive response regarding the revision of our manuscript. We have now taken into account the editorial considerations and incorporated the final revisions suggested by the reviewers. The following details have been amended:

Editorial considerations

- We have amended the manuscript title as requested.
- We are of course happy to publish the study as an article rather than a report. However, we seem to be unable to change the manuscript type during the submission process.
- We have added a Table of Contents to the Appendix.
- We have re-named the EV tables and now cite them as Dataset EV1, EV2 and EV3 in the manuscript text. We also added README.txt files describing each dataset and provide the Dataset - Readme pairs as three separate zip folders.
- We are happy with the amended synopsis text and have re-submitted the suggested version.
- We have modified the synopsis image slightly and re-submitted it as 550 x 250 pixel JPEG as required. We also provide a simplified, smaller version (211 x 157 pixels) for use as a thumbnail.

Please also note that during the re-submission process we tried to edit the author's information for Georg Kustatscher, but were unable to save these changes due to an apparent software bug. In particular, we wanted to add his ORCID ID (0000-0001-8955-0866) and tick the box marking him as co-corresponding author.

Reviewers' comments

Reviewer #1: I liked this revised version very much. The major problems that I saw before were of interpretation (e.g. there is no evidence for selection buffering). The revision places the work better in to context. I also very much liked the new analysis of the stages of buffering which is helpful. In terms of references relating insertion site to gene expression they team should mention:

Akhtar W, de Jong J, Pindyurin AV, Pagie L, Meuleman W, de Ridder J, Berns A, Wessels LF, van Lohuizen M,, van Steensel B. Chromatin position effects assayed by thousands of reporters integrated in parallel (2013). *Cell* 154(4):914-27

This finds evidence that transgenes adopt the local expression profile and can sometimes influence the expression of neighbours. A very nice experiment.

We have added the following sentence to the penultimate abstract of the Introduction section (page 4): "Transgenes can also affect the mRNA expression levels of endogenous genes located close to the insertion site (Akhtar et al. 2013)."

Reviewer #2: The revised manuscript has detailed response in the revised version. The authors added a lot more results to better support the main conclusion of the paper. In general the revised manuscript should be accepted for publication but with one point should be revised:

On page 12, "Buffering of non-functional mRNA co-expression is a non-selective process" The title of this section seems over-claimed. I appreciate that the authors used several posttranscriptional control processes to support that functional-related co-expressed mRNA sustained at the protein level, but claiming that "non-functional mRNA co-expression is a non-selective process" seems to have no support in terms of direct evidence. In particular, the authors do not discuss any potential "selective process that specifically targets closeby genes and disentangles their expression patterns".

It seems more appropriate to claim rather the "buffering of non-functional mRNA co-expression tends to be a non-selective process". Other related claims in this section and the Discussion should be revised as well.

We have softened our conclusions as suggested, in the title as well as the body of this paragraph (page 13). There was no need to update the Discussion section, as it does not go into detail regarding the selective vs non-selective nature of the buffering mechanism. In the abstract we already had a more carefully worded explanation ("buffering *likely* reflects a lack of coordination of post-transcriptional regulation").

With these changes we hope you will find our manuscript acceptable for publication. Please don't hesitate to contact us should you require any further information or modifications.

Thank you,

Kind regards,

Juri Rappsilber

Thank you again for sending us your revised manuscript. We are now satisfied with the modifications made and I am pleased to inform you that your paper has been accepted for publication.

Corresponding Author Name: Georg Kustatscher and Juri Rappsilber

Manuscript Number: MSB-17-7548